# Benchmarking joint multi-omics dimensionality reduction approaches for the study of cancer

Laura Cantini [1✉], Pooya Zakeri [2,5], Celine Hernandez [1,6], Aurelien Naldi [1,7], Denis Thieffry [1], Elisabeth Remy [3] & Anaïs Baudot [2,4✉]

High-dimensional multi-omics data are now standard in biology. They can greatly enhance our understanding of biological systems when effectively integrated. To achieve proper integration, joint Dimensionality Reduction (jDR) methods are among the most efficient approaches. However, several jDR methods are available, urging the need for a comprehensive benchmark with practical guidelines. We perform a systematic evaluation of nine representative jDR methods using three complementary benchmarks. First, we evaluate their performances in retrieving ground-truth sample clustering from simulated multi-omics datasets. Second, we use TCGA cancer data to assess their strengths in predicting survival, clinical annotations and known pathways/biological processes. Finally, we assess their classification of multi-omics single-cell data. From these in-depth comparisons, we observe that intNMF performs best in clustering, while MCIA offers an effective behavior across many contexts. The code developed for this benchmark study is implemented in a Jupyter notebook—multi-omics mix (momix)—to foster reproducibility, and support users and future developers.

[1] Computational Systems Biology Team, Institut de Biologie de l'Ecole Normale Supérieure, CNRS, INSERM, Ecole Normale Supérieure, Université PSL, 75005 Paris, France. [2] Aix Marseille Univ, INSERM, MMG, Marseille Medical Genetics, CNRS, Turing Center for Living Systems, Marseille, France. [3] Aix Marseille Univ, CNRS, Centrale Marseille, I2M, Turing Center for Living Systems, Marseille, France. [4] Barcelona Supercomputing Center (BSC), Barcelona 08034, Spain. [5] Present address: Centre for Brain and Disease Research, Flanders Institute for Biotechnology (VIB), Leuven, Belgium and Department of Neurosciences and Leuven Brain Institute, KU Leuven, Leuven, Belgium. [6] Present address: Université Paris-Saclay, CEA, CNRS, Institute for Integrative Biology of the Cell (I2BC), 91198 Gif-sur-Yvette, France. [7] Present address: Inria Saclay Ile de France, EP Lifeware, Palaiseau, France. ✉email: laura.cantini@ens.fr; anais.baudot@univ-amu.fr

D ue to the advent of high-throughput technologies, high-dimensional "omics" data are produced at an increasing pace. In cancer biology, in particular, national and international consortia, such as The Cancer Genome Atlas (TCGA), have profiled thousands of tumor samples for multiple molecular assays, including mRNA, microRNAs, DNA methylation, and proteomics[1]. Moreover, multiomics profiling approaches are currently being transposed at single-cell level, which further stresses the need for methods and tools enabling the joint analysis of such large and diverse datasets[2].

While multiomics data are becoming more accessible, studies combining different omics are more common. This multiomics integration is frequently performed by sequentially combining results obtained on single omics (a.k.a. late or early integration), but the genuine joint analysis of multiomics data (a.k.a. intermediate integration) remains very rare[3]. Achieving proper multiomics integration is crucial to bridge the gap between the vast amount of available omics and our current understanding of biology. By integrating multiple sources of omics data, we can reduce the effect of experimental and biological noise. In addition, different omics technologies are expected to capture different aspects of cellular functioning. Indeed, the different omics are complementary, each omics containing information that is not present in others, and multiomics integration is thereby expected to provide a more comprehensive overview of the biological system. In cancer research, omics have been profiled at different molecular layers, such as genome, transcriptome, epigenome, and proteome. Integrating these large-scale and heterogeneous sources of data allows researchers to address crucial objectives, including (i) classifying cancer samples into subtypes, (ii) predicting the survival and therapeutic outcome of these subtypes, and (iii) understanding the underlying molecular mechanisms that span through different molecular layers[4].

Designing theoretical and computational approaches for the joint analysis of multiomics datasets is currently one of the most relevant and challenging questions in computational biology[4,5]. Indeed, the different types of omics have a large number of heterogeneous biological variables and a relatively low number of biological samples, thereby inducing statistical and computational challenges, in addition to the typical challenges of "Big Data". Moreover, each omics has its own technological limits, noise, and range of variability. All these elements can mask the underlying biological signals. Multiomics integrative approaches should be able to capture not only signals shared by all omics data, but also those emerging from the complementarity of the various omics data.

The joint analysis of multiple omics can be performed with various integrative approaches, classified in broad categories[5,6]. Bayesian methods, such as Bayesian Consensus Clustering (BCC)[7], build a statistical model by making assumptions on data distribution and dependencies. Network-based methods, such as Similarity Network Fusion (SNF)[8], infer relations between samples or features in each omics layer, and further combine the resulting networks. Dimensionality Reduction (DR) approaches decompose the omics into a shared low-dimensional latent space[9,10]. Four recent reviews tested and discussed some of these methods from a clustering performance perspective[11–14]. Pierre-Jean et al.[13], Rappoport et al.[11], and Tini et al.[14] selected one method from each of the aforementioned three categories, while Chauvel et al.[12] focused on Bayesian and DR approaches.

From these initial reviews, DR approaches emerged as particularly well-performing. Of note, DR is employed in computational biology in different contexts, such as data visualization or matrix completion. We focus here on DR for multiomics data integration. DR methods are well-adapted to solve high-dimensional mathematical problems. Furthermore, the richness of the information contained in their output enhances their relevance for multiomics integration. Indeed, DR methods enable the classification of samples (clustering/subtyping), the clinical characterization of the identified clusters/subtypes and a variety of other downstream analyses, including the analysis of cellular processes and/or pathways (Fig. 1a). Thus, DR combined with dedicated downstream analyses provides information on all the key objectives mentioned above, namely the classification of samples into subtypes, their association with outcome/survival, as well as the reconstruction of their underlying molecular mechanisms. As a consequence, the design of DR approaches for the joint analysis and integration of multiple omics (jDR) is currently a highly active area of research[8,9,11,12,15].

Here, we report an in-depth comparison of nine representative state-of-the-art multiomics joint Dimension Reduction (jDR) approaches, in the context of cancer data analysis. We extensively benchmark these approaches, spanning the main mathematical formulations of multiomics jDR, in three different contexts (Fig. 1b). First, we simulate multiomics datasets and evaluate the performance of the nine jDR approaches in retrieving ground-truth sample clustering. Second, we use TCGA multiomics cancer data to assess the strengths of jDR methods in predicting survival, clinical annotations, and known pathways/biological processes. Finally, we evaluate the performance of the methods in classifying multiomics single-cell data from cancer cell lines.

All these analyses allow formulating recommendations and guidelines for users, as well as indications for methodological improvements for developers. We also provide the Jupyter notebook multiomics mix (momix) and its associated Conda environment containing all the required libraries installed (ComputationalSystemsBiology/momix-notebook). Overall, momix can be used to reproduce the benchmark, but also to test jDR algorithms on other datasets, and to evaluate novel jDR methods and compare them to reference ones.

## Results

**Joint Dimensionality Reduction approaches and principles.** Joint Dimensionality Reduction (jDR) approaches aim to reduce high-dimensional omics data into a lower dimensional space. The rationale behind the use of jDR in biology is that the state of a biological sample is determined by multiple concurrent biological factors, from generic processes (e.g., proliferation and inflammation) to cell-specific processes. When measuring omics data, we take a snapshot of the state of a biological sample and thus detect a convoluted mixture of various biological signals active in the sample. The goal of jDR is to deconvolute this mixture and expose the different biological signals contributing to the state of the biological sample. We consider $P$ omics matrices $\mathbf{X}^i$, $i = 1, ..., P$ of dimension $n_i \times m$ with $n_i$ features (e.g., genes, miRNAs, CpGs) and $m$ samples. A jDR jointly decomposes the P omics matrices into the product of $n_i \times k$ omics-specific *weight/projection matrices* ($\mathbf{A}^i$) and a $k \times m$ *factor matrix* ($\mathbf{F}$) (Fig. 1a). Here and in the following, we will denote as factors the rows of the factor matrix, and as *metagenes* the columns of the weight/projection matrix corresponding to transcriptomic data (see Methods). Factors and metagenes represent the projections on the sample space and gene space, respectively, of the biological signals present in the profiled samples. The factor matrix ($\mathbf{F}$) can be used to cluster samples, while the columns of the weight matrices ($\mathbf{A}^i$) can be used to extract markers by selecting the top-ranked genes, or to identify pathways by applying preranked GSEA (see ref. [10] for further details). A description of the mathematical formulations of the nine jDR approaches is provided in the Methods section.

Various methods exist to perform jDR (Supplementary Table 1). These methods are based on different underlying

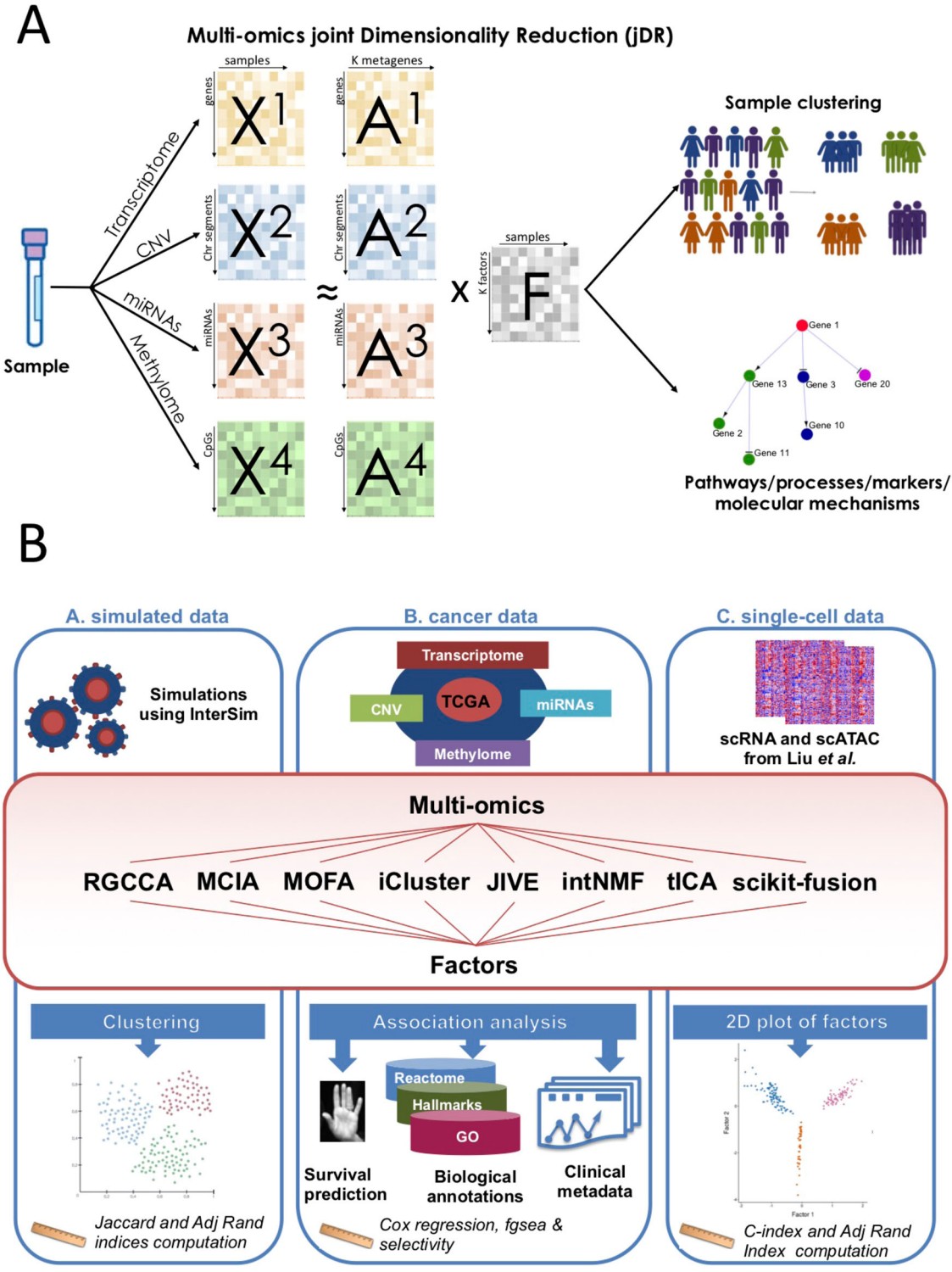

**Fig. 1 Joint Dimensionality Reduction methods and benchmark workflow overview. a** Multiomics are profiled from the same sample. Each omics corresponds to a different matrix $X^i$. jDR methods factorize the matrices $X^i$ into the product of a factor matrix F and weight matrices $A^i$. These matrices can then be used to cluster samples and identify molecular processes. **b** Workflow of our benchmark, subdivided in three subparts: First, we simulated multiomics datasets and evaluated the performance of the nine jDR approaches in retrieving ground-truth sample clustering. Second, we used TCGA multiomics cancer data to assess the strengths of jDR methods in predicting survival, clinical annotations, and known pathways/biological processes. Finally, we evaluated the performances of the methods in classifying multiomics single-cell data from cancer cell lines.

mathematical formulations, including Principal Components Analysis, Factor analysis, co-inertia analysis, Gaussian latent model, matrix-tri-factorization, Non-negative Matrix Factorization, CCA or tensor representations. We selected nine jDR approaches representative of these main mathematical formulations (Fig. 2), focusing on methods able to combine more than two omics, implemented in R or Python, and with software readily available and documented. These jDR approaches are

| jDR approach | Underlying approach | Constraints on the factors | Features or samples matching requirements | Implementation | Simulated data sub-benchmark (percentage of cancer) | Cancer sub-benchmark: survival (percentage of cancers) | Cancer sub-benchmark: clinical annotations (percentage of cancers) | Cancer sub-benchmark: biological annotations (percentage of cancers) | Single-cell sub-benchmark (1 - C-index) |
|---|---|---|---|---|---|---|---|---|---|
| RGCCA | Canonical Correlation Analysis (CCA) | omics-specific factors | matching of samples | R package RGCCA | 83% | 70% | 60% | 60% | 0.92 |
| MCIA | Co-Inertia Analysis (CIA) | omics-specific factors | matching of samples | R package omicade4 | 100% | 70% | 50% | 80% | 0.98 |
| MOFA | Factor Analysis (FA) (Bayesian) | mixed factors | matching of samples (partial matching allowed) | R code on github bioFAM/MOFA | 100% | 50% | 50% | 50% | 0.88 |
| MSFA | Factor Analysis (FA) (Bayesian) | mixed factors | matching of samples | R code on github rdevito/MSFA | 0% | NA | NA | NA | 1.00 |
| intNMF | Non-Negative Matrix Factorization (NMF) | shared factors | matching of samples | R package intNMF | 100% | 10% | 20% | 50% | 0.97 |
| iCluster | Gaussian latent variable model | shared factors | matching of samples | R package iCluster | 100% | 40% | 30% | 60% | 0.75 |
| JIVE | Principal Component Analysis (PCA) | mixed factors | matching of samples (partial matching allowed) | R package r.jive | 0% | 70% | 40% | 40% | 0.72 |
| tICA | Independent Component Analysis (ICA) | shared factors | matching of both samples and features (tensor) | R scripts in supplementary material of Teschendorff et al. | 0% | 50% | 20% | 90% | 1.00 |
| Scikit-fusion | Matrix tri-factorization | shared factors | matching of samples | Python scripts on github marinkaz/scikit-fusion | 17% | 60% | 20% | 60% | 0.81 |

**Fig. 2 Dimensionality reduction approaches benchmarked in this study.** The list of the jDR methods benchmarked in this study is reported together with their underlying approach, constraints on the factors, features or samples matching requirements, implementation and a summary of the benchmarking performances. The benchmarking performances are organized as follows: simulated data, cancer survival, cancer clinical annotations, biological annotations, and single cell.

iCluster[16], Integrative NMF (intNMF)[17], Joint and Individual Variation Explained (JIVE)[18], Multiple co-inertia analysis (MCIA)[19], Multi-Omics Factor Analysis (MOFA)[15], Multi-Study Factor Analysis (MSFA)[20], Regularized Generalized Canonical Correlation Analysis (RGCCA)[21], matrix-tri-factorization (scikit-fusion)[22], and tensorial Independent Component Analysis (tICA)[23].

Seven of the nine jDR approaches are extensions of DR methods previously used for single-omics datasets: intNMF is an extension of non-Negative Matrix Factorization (NMF); tICA is an extension of Independent Component Analysis (ICA); MCIA and JIVE are different extensions of Principal Component Analysis (PCA); and MOFA, MSFA, and iCluster are extensions of Factor Analysis. As a consequence, the different jDR algorithms make different assumptions on the distribution of the factors (Methods). The different jDR approaches also make different assumptions on the across-omics constraints on the factors (Fig. 2). Some algorithms, such as intNMF, consider the factors to be *shared* across all omics datasets. In contrast, the factors of RGCCA and MCIA are different for each omics layer, i.e., they are omics-specific factors. These omics-specific approaches still maximize some measures of interrelation between the omics-specific factors, such as their correlation (RGCCA), or their co-inertia (MCIA). Finally, JIVE and MSFA consider mixed factors, decomposing the omics data as the sum of two factorizations, one containing a unique factor matrix shared across all omics, and the second having omics-specific factor matrices. Of note, jDR methods considering omics-specific and mixed factors take into account the complementarity of the multiomics data.

Most of the jDR approaches can manage different features (e.g., genes, miRNAs, CpGs…), but require a match between the samples of the different omics datasets (columns of the $\mathbf{X^i}$ matrices, see Fig. 2). Some algorithms, such as MOFA, scikit-fusion and JIVE, can also cope with omics matrices having not all samples in common. This is particularly suitable for multiomics integration, as missing samples are frequent in data collections, such as in TCGA. For the sake of comparison, we applied here all methods considering only the samples profiled for all omics. Tensorial approaches, represented by tICA, require by definition that all matrices $\mathbf{X^i}$ have exactly the same samples and features. Nonetheless, the features of multiomics data are frequently different (e.g., genes, miRNAs). A possible strategy to have the same features for all omics would be to convert all the features to the same level, e.g., gene symbols. This is sometimes unfeasible: miRNAs cannot be converted to gene symbols, for instance. We applied here another strategy, where we considered for each omics the matrix of correlation-of-correlation between samples (Methods). Both strategies imply a loss of information, which can

affect the results of the omics integration. In addition, the number of features $n_i$ in the various omics is highly variable, going for instance in TCGA from 800 microRNAs to 5000 CpGs to 20.000 genes. Omics containing more features will have a higher weight in the jDR output. To overcome this issue, in the following, we will first select features based on their variability, and thus make the number of features of the various omics comparable.

Noteworthy, intNMF and iCluster produce, in addition to the factors, a clustering of the samples. Scikit-fusion can combine omics data with additional annotation (i.e., side information, such as pathway or process annotations). However, for the sake of comparisons with other algorithms, scikit-fusion is applied here without side information.

**Benchmarking joint Dimensionality Reduction approaches on simulated omics datasets**. We first evaluated the jDR approaches on artificial multiomics datasets (Fig. 3a). We simulated these datasets using the *InterSIM* CRAN package[24]. This package generates three omics datasets with imposed reference clustering. Starting from real omics (DNA methylation, transcriptome and protein expression) extracted from TCGA ovarian cancer datasets, InterSIM generates clusters and associates features to these clusters by shifting their mean values by a fixed amount. InterSIM preserves the covariance matrix between all pairs of omics and thereby maintains realistic inter- and intraomics relationships. Importantly, we selected this approach to avoid making assumptions on the distribution of the data. Indeed, alternative simulation approaches assume specific sample distributions (e.g., Gaussian in ref. [14]). Assuming a Gaussian distribution, for instance, would favor jDR methods that also make the assumption of Gaussian sample distribution.

We simulated multiomics data with five, ten, and fifteen clusters. In addition, each set of clusters is simulated in two versions, either with all clusters of the same size, or with clusters of variable random sizes (Methods).

We applied the nine jDR methods, requiring the decomposition of multiomics data into five, ten, and fifteen factors, depending on the simulated datasets. The performances of the nine jDR approaches are then compared based on their clustering of samples. As mentioned before, intNMF and iCluster are intrinsically designed for sample clustering, while the remaining seven algorithms detect factors without providing a direct clustering. Accordingly, we applied directly intNMF and iCluster. For the seven other algorithms, we obtained the clustering of the samples by applying k-means consensus clustering to the factor matrix (Methods).

The agreement between the clustering obtained with the various jDR algorithms and the ground-truth clustering is measured with

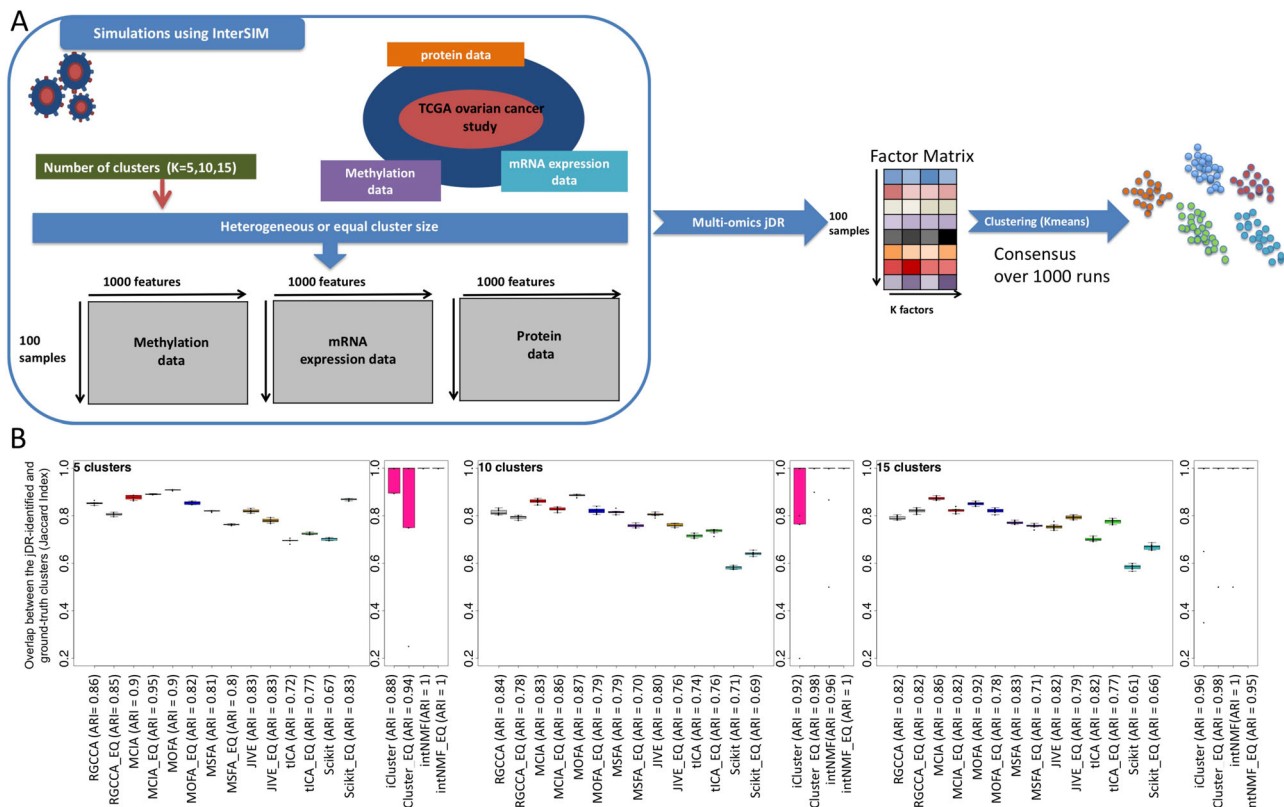

**Fig. 3 jDR clustering of simulated multiomics datasets. a** Workflow of the simulation sub-benchmark from the data generation with interSIM, to the jDR output and its clustering based on k-means. **b** Boxplots of the Jaccard Index computed between the clusters identified by the different jDR methods and the ground-truth clusters imposed on the simulated data (for 5, 10, and 15 imposed clusters). For each method (e.g., RGCCA), performances on heterogeneous and equally sized clusters are reported (denoted as RGCCA and RGCCA_EQ, respectively). The corresponding Adjusted Rand Index (ARI) values are further reported near to the name of the jDR methods along the x-axis. The number of samples here considered is 100 and the results are obtained over 1000 independent runs of k-means clustering. Data are presented as mean values ± sd, whiskers denote max, and min values.

the Jaccard Index (JI) and Adjusted Rand Index (ARI) (Methods). First, we observed that all methods perform reasonably well in the different simulated scenarios (JIs > =0.6, ARI > =0.6, Fig. 3b). The two algorithms intrinsically designed for clustering, namely intNMF and iCluster, display the best performances. In particular, intNMF retrieves perfectly the ground-truth clusters (JI ~ 1, 0.9 < ARI < =1). iCluster presents some variability for five and ten clusters, independently of the size distribution of the clusters. Regarding the remaining seven jDR approaches, MCIA, MOFA, and RGCCA are overall the best-performing methods. These methods are indeed among the top-three best algorithms in 6/6, 6/6, and 5/6 simulated scenarios, according to JI, and in 6/6, 3/6, and 5/6, according to ARI, respectively. tICA and scikit-fusion are the less effective methods in this benchmark. tICA structures the multiomics data into a tensor. As described previously, to obtain these tensors, we transformed the omics data into correlation-of-correlation matrices, which might induce a loss of information. scikit-fusion is designed to work with side information, which is used to build a relation network connecting the various entities (e.g., samples, genes, proteins). However, for the sake of comparison with the other jDR methods, side information was not considered, and this could have affected the results of the algorithm.

**Benchmarking joint Dimensionality Reduction approaches on cancer datasets.** In the second step, we downloaded TCGA multiomics data for ten different cancer types[11] (http://acgt.cs.tau.ac.il/multi_omic_benchmark/download.html). These data are composed of three omics layers: gene expression, DNA methylation, and miRNA expression. The number of samples ranges from 170 for

Acute Myeloid Leukemia (AML) to 621 for Breast cancer. Importantly, we do not have ground-truth cancer subtypes to evaluate the performances of the jDR methods. However, in Breast cancer, we compared the jDR clustering results with two subtypings: the ER/PR/HER-2 subtyping based on Estrogen Receptor (ER), Progesterone Receptor (PR) and HER-2 immunohistochemistry markers[25], and the Cluster of Cluster Assignment (COCA) integrative classification performed by the TCGA consortium[26]. Both subtypings cannot be considered as ground-truth for evaluating jDR clustering performances. The ER/PR/HER-2 overlaps with the PAM50 subtyping, which is obtained using only transcriptomics data, and composed of four subtypes: Basal, Her2, Luminal A and Luminal B[27]. The COCA subtyping is integrative but has been obtained by separately clustering different omics and then performing a consensus of the obtained results. Thereby, it does not take into account the complementarity of the various omics.

We decomposed the multiomics Breast cancer datasets in four factors, and used the Jaccard Index (JI) and Adjusted Rand Index (ARI) to evaluate the overlap between the clustering obtained from these four factors and the Breast cancer subtypings ER/PR/HER-2 and COCA (Supplementary Fig. 1). Most of the methods display low JI ([0.2;0.6]) and ARI ([0.2;0.5]) values. JIVE shows the best performances according to both JI and ARI (JI = 0.4, ARI = 0.4). MCIA has the best performances according to ARI (ARI = 0.5), and intNMF has good performances according to JI, but with high variability ([0.2;0.8]), which results in a low ARI value (ARI = 0.28).

In order to evaluate the methods on the full set of cancer multiomics, we then tested the associations of the factors with

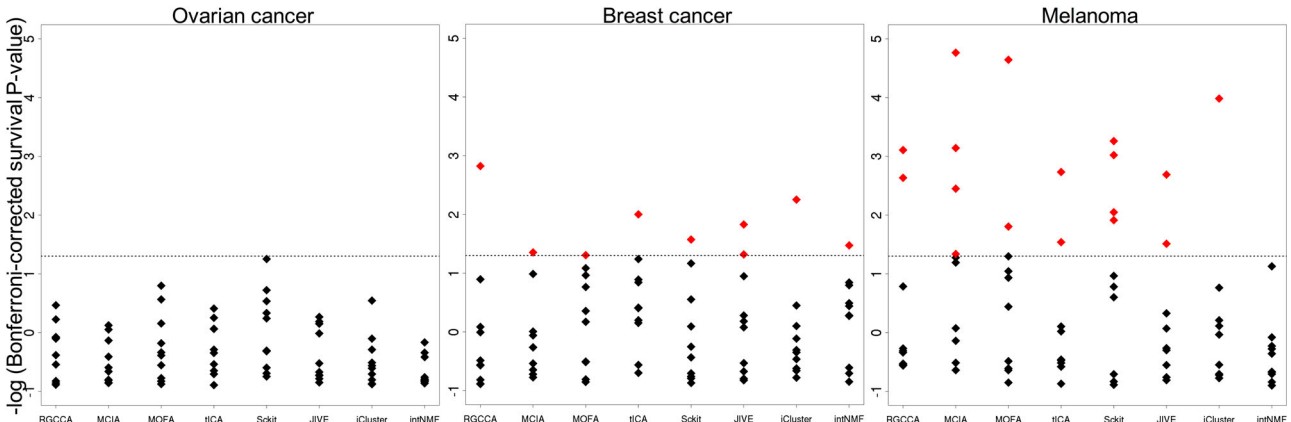

**Fig. 4 graphic summary of the cancer sub-benchmark. a** Testing the association of jDR factors with survival; **b** Testing the association of jDR factors with clinical annotations; **c** Graphical explanation of the selectivity score: measuring the one-to-one mapping between factors and clinical/biological annotations; **d** Testing association of jDR factors with biological processes and pathways.

**Fig. 5 Identification of factors predictive of survival in ovarian, breast, and melanoma cancer samples by the jDR methods.** For each method the Bonferroni-corrected p-values associating each of the 10 factors to survival (Cox regression-based survival analysis) are reported. The dot lines correspond to a corrected *p*-value threshold of 0.05. The results corresponding to the other seven cancer types are presented in Supplementary Fig. 1A.

survival, clinical annotations and biological annotations (Fig. 4). It is to note that the Factor Analysis approach MSFA did not converge to any solution and was thereby not considered. We applied the eight remaining jDR approaches to each of the ten cancer multiomics datasets, jointly decomposing them in ten factors, as in the work of Bismeijer and colleagues[28]. Most cancer subtyping approaches indeed revealed ten or fewer clusters of samples (i.e., subtypes). We hence compared the performances of the eight jDR algorithms regarding their ability to identify factors predictive of survival, as well as factors associated with clinical annotations. We also evaluated the weight matrices resulting from the jDR methods by assessing their enrichment in known biological pathways and processes.

To test the association of the jDR factors with survival, we used the Cox proportional-hazards regression model (Fig. 4a). We

observed first that the number of factors associated with survival depends more on the cancer type than on the jDR algorithm (Fig. 5 and Supplementary Fig. 2). Indeed, for three cancer types (Colon, Lung, and Ovarian), none of the jDR methods was able to identify survival-associated factors. This result is in agreement with previous observations testing the association of multiomics clusters with survival on the same TCGA data with the log-rank test[11]. In four other cancer types (sarcoma, liver, kidney, and breast), all jDR algorithms identified one or two survival-associated factors. Finally, in Melanoma, GBM, and AML, the majority of the jDR methods identified three or four survival-associated factors. In general, MCIA, RGCCA, and JIVE achieved the best performances, finding factors significantly associated with survival in seven out of ten cancer types. These approaches also offered the most significant p-values in the higher number of cancer types. MCIA performed the best for kidney cancer

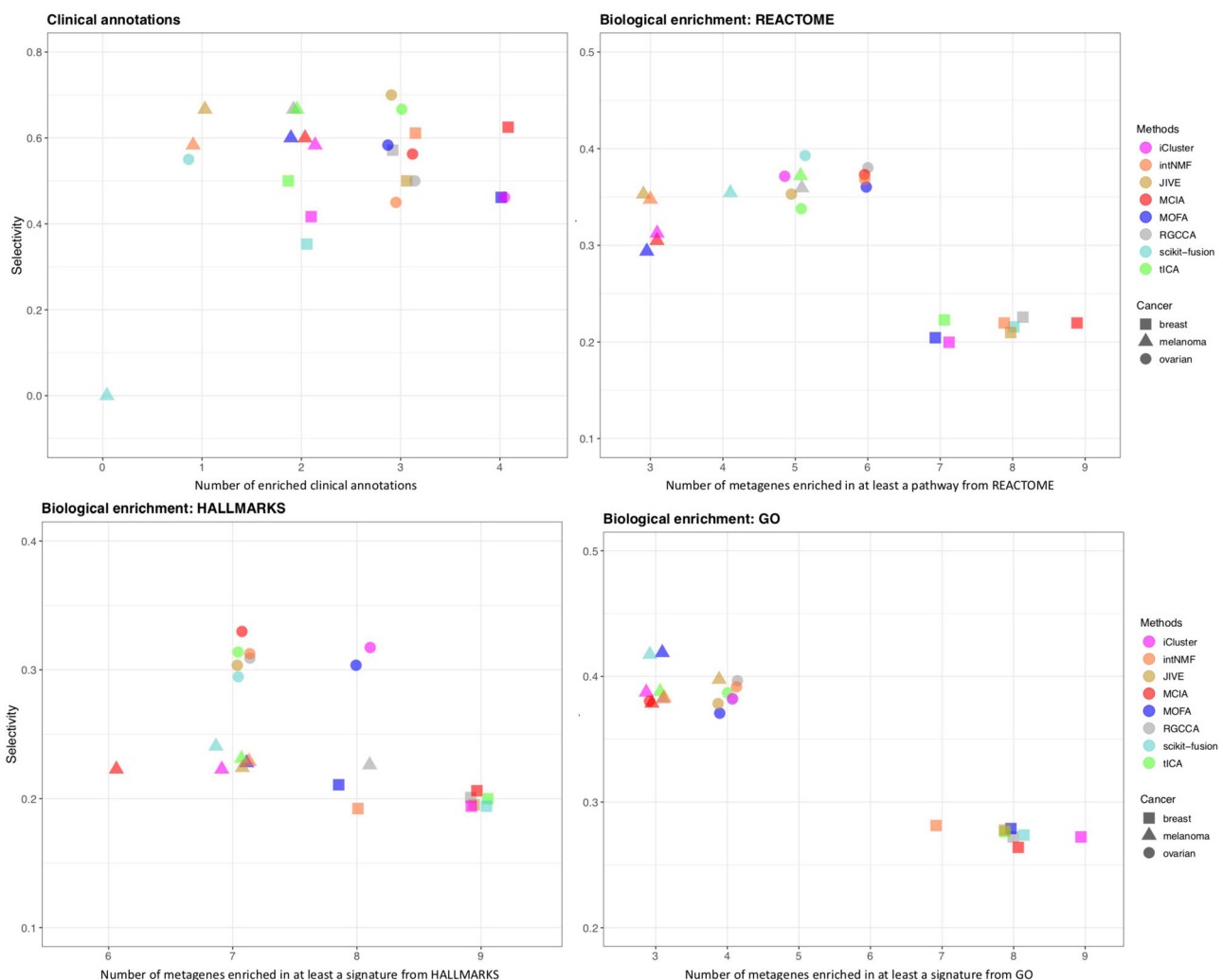

**Fig. 6 Identification of factors associated with clinical annotations, and metagenes associated with biological annotations in ovarian, breast, and melanoma samples, by the jDR methods.** For clinical annotations, the plot represents, for each method, the number of clinical annotations enriched in at least one factor together with the selectivity of the associations between the factors and the clinical annotations (Method). For the three annotation sources (MsigDB Hallmarks, REACTOME and Gene Ontology), the number of metagenes identified by the different jDR methods enriched in at least a biological annotation are plotted against the selectivity of the associations between the metagene and the annotation. See Supplementary Fig. 3 for the results corresponding to the other seven cancer types.

(corrected $p$-value $= 10^{-4}$), sarcoma ($10^{-5}$), and melanoma ($10^{-5}$); JIVE performed the best in AML ($10^{-3}$) and liver cancer ($10^{-4}$); and RGCCA performed the best in Breast cancer ($10^{-3}$). Furthermore, RGCCA, MCIA, and JIVE showed the most promising results for the cancer types having overall less survival-associated factors (Sarcoma, Liver, Kidney, and Breast, Fig. 5 and Supplementary Fig. 2). It is to note that these jDR methods are also the best performing when compared to DR applied to transcriptome alone (Supplementary Fig. 3).

Afterward, we assessed the association of the jDR factors with clinical annotations (see Fig. 4b for the methodology, Fig. 6 and Supplementary Fig. 4 for the results). We selected four clinical annotations: "age of patients," "days to new tumor," "gender", and "neo-adjuvant therapy administration" (Methods). To test the significance of the associations of the factors identified by the jDR methods with these clinical annotations, we used Kruskal–Wallis tests for multi-class annotations ("age of patients" and "days to new tumor"), and Wilcoxon rank-sum for binary annotations ("gender" and "neo-adjuvant therapy administration"). In addition, we intended to evaluate the methods not only by their capacity to associate factors with clinical

annotations, but also by their ability to achieve these associations with a one-to-one mapping between a factor and a clinical annotation, i.e., their selectivity (Fig. 4c). Indeed, a jDR method detecting one factor associated with multiple clinical annotations cannot distinguish the annotations from each other. To the contrary, a jDR method detecting multiple factors associated with only one clinical annotation does not maximally explore the spectrum of all possible annotations. We defined a selectivity score having a maximum value of 1 when each factor is associated with one and only one clinical annotation, and a minimum of 0 when all factors are associated with all clinical annotations (Methods). The average selectivity value of all methods across all cancer types is 0.49. The top methods in each cancer type are defined as those having a maximum number of factors associated with clinical annotations, together with a selectivity value above the average. RGCCA, MCIA, and MOFA are overall the best-performing algorithms, since they rank among the top three methods in 6/10, 5/10, and 5/10 cancer types, respectively. In contrast, intNMF, scikit-fusion, and tICA are less effective (among the top three methods in only two out of ten cancer types).

Finally, we assessed the jDR methods performances in associating factors with biological processes and pathways (see Fig. 4d for the methodology, Fig. 6 and Supplementary Fig. 4 for the results). To achieve this goal, we needed to take into account genes (i.e., weight matrices) and not samples (i.e., factor matrices). We computed the number of metagenes (corresponding to the columns of the transcriptomics weight matrix) enriched in at least one biological annotation from Reactome, Gene Ontology (GO), and cancer Hallmarks annotation databases (Methods). An optimal jDR method should maximize the number of metagenes enriched in at least one biological annotation, while optimizing also the selectivity (defined as above for clinical annotations and in the Methods). The average selectivity of all methods across the ten cancers is 0.3 for Reactome, 0.35 for GO, and 0.26 for cancer Hallmarks. The top methods in each cancer type are defined as those having a maximum number of metagenes associated with biological annotations, together with selectivity values above the average. Scikit-fusion, tICA, and RGCCA are overall the best-performing algorithms for Reactome annotations (ranking among the top three methods in 4/10, 3/10, 3/10 cancer types, respectively). tICA, iCluster and MCIA provided the best performances in cancer Hallmarks annotations (ranked among the top three methods in 4/10, 3/10, 3/10 cancers, respectively) and MCIA, intNMF and iCluster performed the best in GO annotations (ranked among the top three methods in 4/10, 3/10, 3/10 cancers, respectively). Overall, among all jDR methods, tICA and MCIA resulted in the most promising results for two out of three annotation databases considered in this study, and displayed the best average performances across the three annotations databases (Fig. 6).

**Benchmarking joint Dimensionality Reduction approaches on single-cell datasets.** Similarly to bulk multiomics analyses, the joint analysis of single-cell multiomics is expected to provide tremendous power to untangle the cellular complexity. jDR approaches are expected to compensate for the strong intrinsic limitations of single-cell multiomics, such as small number of sequencing reads, systematic noise due to the stochasticity of gene expression at single-cell level, or dropouts[29–31]. However, the nine jDR algorithms that we are considering (except MOFA) have been designed and applied to bulk multiomics data. It is therefore crucial to evaluate and benchmark the performances of these jDR algorithms for single-cell multiomics integration.

To test the jDR approaches on single-cell omics, we fetched scRNA-seq and scATAC-seq, simultaneously measuring gene expression and chromatin accessibility on three cancer cell lines (HTC, Hela and K562) for a total of 206 cells, and reported in the study of Liu and colleagues[32]. As these cells have been obtained from three different cancer cell lines, we expect that the first two factors of the various jDR approaches would cluster single-cells according to their cancer cell line of origin.

The first two factors of the nine jDR algorithms show overall good performances to separate cells according to cell lines of origin (Fig. 7a). To compare quantitatively these results, we measured the C-index with values in the range [0,1], where 0 represents an optimal clustering (Methods). According to our results, tICA and MSFA are best-performing jDR methods with a C-index of 0, immediately followed by MCIA and intNMF (C-indices 0.018 and 0.025, respectively), followed by RGCCA, MOFA, and scikit-fusion (C-indices 0.077, 0.12, 0.19, respectively), and finally, JIVE and iCluster (C-indices 0.23 and 0.25, respectively). To further compare the performances of jDR approaches with state-of-the-art single-cell multiomics integrative tools, we further included in our analysis Seurat[33] and LIGER[34]. Importantly, Seurat does not output factors as the other methods. We thus compared the methods based on their clustering abilities

following the same procedure as in the simulation benchmark (Methods). Strikingly, although initially not designed for single-cell data analysis, jDR methods perform equally well or better than Seurat and LIGER (Fig. 7b). Overall, with MCIA, tICA and MSFA were the best-performing algorithms.

**Multiomics mix (momix) Jupyter notebook.** To foster the reproducibility of all the results and figures presented in this benchmark study, we implemented the corresponding code in a Jupyter notebook available at https://github.com/ComputationalSystemsBiology/momix-notebook, together with a Conda environment containing all the required libraries installed. Written in R, this notebook is structured in three main parts corresponding to the three test cases here considered (simulated data, bulk TCGA cancer data and single-cell data). Importantly, this notebook can be easily modified to test the nine jDR algorithms on user-provided datasets. The notebook can also be easily extended to benchmark novel jDR algorithms on our three test cases. Full documentation to achieve these goals is included in the notebook.

**Discussion**
We benchmarked in-depth nine jDR algorithms, representative of multiomics integration approaches, in the context of cancer data analysis. In contrast to previous comparisons[11–14], our benchmark not only focuses on the evaluation of the clustering outputs, but also evaluates the biological, clinical, and survival annotations of the factors and metagenes. Existing comparisons also mainly use simulated data, while we here further consider large datasets of bulk cancer multiomics, as well as single-cell data.

When performing clustering on simulated multiomics datasets, intNMF, intrinsically designed as a clustering algorithm, displayed the most promising results. MCIA, MOFA, and RGCCA showed the best performance among the set of methods not intrinsically designed for clustering. In the cancer data benchmark, when we evaluated the associations of the factors with survival or clinical annotations, MCIA, JIVE, MOFA, and RGCCA were the most efficient methods. When assessing the associations of the metagenes with biological annotations, MCIA and tICA were the most efficient. Finally, in the last benchmark, when clustering single-cell multiomics data, MSFA and tICA, as well as MCIA and intNMF, outperformed other approaches.

As mentioned earlier, intNMF, representative of the Non-negative Matrix Factorization (NMF) approaches, performs well for the clustering tasks, i.e., for detecting substantial patterns of variation across the omics datasets. This is observed for both simulated bulk data clustering and single-cell data clustering. Hence, intNMF should be prioritized by researchers focusing on clustering samples. However, intNMF is not effective when assessing the quality of individual factors and metagenes, as observed in the bulk cancer sub-benchmark. Our results rather suggest that for the study of factor-level information, such as associations with clinical annotation or survival, MCIA, JIVE, MOFA, and RGGCA are the best algorithms to use. When focusing on the underlying biology of the metagenes, tICA and MCIA should be prioritized. Indeed, we showed that these approaches are efficient to detect pathways or processes, but they could also be interesting to identify biomarkers or other molecular mechanisms. Finally, our study highlights the versatility of MCIA. Indeed, MCIA is the method with the most consistent and effective behavior across all the different subbenchmarks. MCIA can thereby be particularly useful for researchers interested in applying jDR with multiple or more open biological questions.

In the future, it would be interesting to extend our benchmark to evaluate jDR methods on discrete omics data. Indeed our current benchmark focuses on continuous data (e.g., expression,

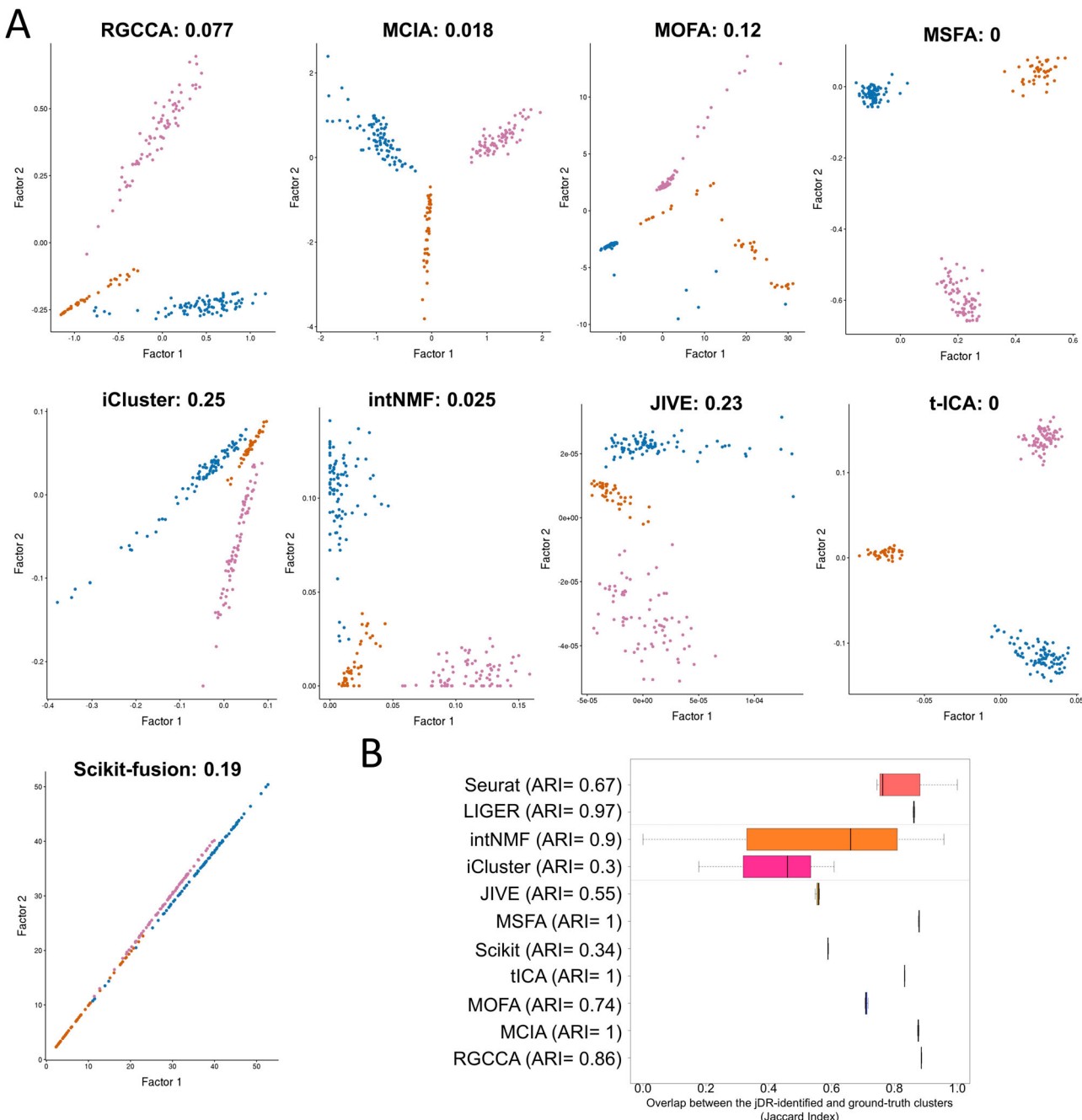

**Fig. 7 jDR clustering of single-cell multiomics according to the cancer cell line of origin. a** Scatterplots of factor 1 and 2 (i.e., the first two columns of the factor matrix) are reported for each jDR method. The colors denote the cancer cell line of origin: pink for K562, orange for Hela and blue for HCT. The C-index (in the range [0–1]) reports the quality of the obtained clusters (0 being the best). **b** Boxplots of the Jaccard Index corresponding to the application of jDR plus LIGER and Seurat for single-cell multiomics clustering. The corresponding Adjusted Rand Index (ARI) values are further reported near to the name of the jDR methods along the x-axis. The number of cells here considered is 206 and the results are obtained over 1000 independent runs of k-means clustering. Data are presented as mean values ± sd, whiskers denote max, and min values.

methylation), whereas many omics and annotations can be formalized as discrete data (e.g., copy number, mutation, drug response). Further extensions of our benchmark could also investigate the impact of different variables on the jDR methods, such as the stability of the methods with respect to variations in the structure of omics data (e.g., imbalance in variability or number of features); or optimal performances according to different combinations of omics data (e.g., are three omics more informative than two?). In addition, to make a fairer comparison, we imposed the same numbers of factors to all of jDR methods, but we could

alternatively use the optimal number of factors directly computed by each method, as in the work of Tini and colleagues[14]. Finally, multiomics data are frequently profiled from different sets of patients/samples, leading to missing data, and further extensions of the benchmark could take this point into account.

From a technical perspective, we observed that the methods that seek for omics-specific factors often led to a better performance than the methods designed for finding shared or mixed factors. We hypothesize that jDRs with omics-specific factors could successfully detect not only biological processes shared

across multiple omics, but also those processes that are complementary in multiple sources of omics data. In addition, when using algorithms producing omics-specific factors, we have only evaluated the transcriptome-associated omics-specific factors (Methods). However, the outputs of these methods can often contain additional relevant information in other omics-specific factors. As an example, in our benchmark, the jDR methods did not retrieve factors significantly associated with survival in ovarian cancer. However, a significant association is retrieved with one miRNA-specific factor of RGCCA. We thus suggest developers to prioritize omics-specific factors for further methodological developments. Accordingly, the use of co-inertia (as implemented in MCIA) appears more efficient to enforce relationships across omics than the use of correlation (as implemented in RGCCA). In addition, there is room for development of approaches managing missing data, as many of the best-performing approaches can work only on omics profiled from the same samples. This is also true for the consideration of discrete data as, among the methods considered here, only MOFA and scikit-fusion have been previously applied to such data. Finally, most of the considered methods detect only linear signals. MOFA is the only algorithm in our benchmark that can also detect slightly nonlinear signals, as shown in ref. [15] for mouse embryonic stem cells. As a result, future developments should be directed towards methods that can capture the nonlinear signals present in the data. Developers could take advantage of the momix Jupyter notebook using it to compare novel methods with established ones.

## Methods

We consider $P$ omics matrices $\mathbf{X^i}$, $i = 1, ..., P$ of dimension $n \times m$, with $n_i$ features (e.g., genes, miRNAs, CpGs) and $m$ samples. A jDR jointly decomposes the P omics matrices into the product of $n_i \times k$ omics-specific *weight/projection matrices* ($\mathbf{A^i}$) and a $k \times m$ *factor matrix* ($\mathbf{F}$) and (Fig. 1). $k$ is thus the number of factors of the decomposition. Here and in the following, we will denote as factors the rows of the factor matrix and as *metagenes* the columns of the weight/projection matrix corresponding to transcriptomic data.

**Presentation of the nine jDR algorithms**. We detail here the nine jDR methods benchmarked in momix. We selected default parameters for each approach. Each method can in principle optimize its number of factors to be detected, but for the sake of comparison, we imposed the same number of factors on all approaches. Please note that we followed the mathematical formulations and notations provided in each corresponding publication.

*Integrative non-negative matrix factorization (intNMF)*. intNMF[17] is one of the numerous generalizations of NMF to multiomics data. The method decomposes each omics matrix $\mathbf{X^i}$ into a product of non-negative matrices: the factor matrix $\mathbf{W}$, and an omics-specific matrix $\mathbf{H^i}$:

$$\mathbf{X^i} = \mathbf{WH^i}, \text{ for } i = 1, ..., P \text{ with } \mathbf{W} \text{ and } \mathbf{H^i} \text{ positive matrices for } i = 1, ..., P. \tag{1}$$

The algorithm minimizes the objective function
$\mathbf{Q} = \min_{WH} \sum_{i=1}^{P} \theta^i \left\| \mathbf{X^i} - \mathbf{WH^i} \right\|$.
Once the matrices $\mathbf{W}$ and $\mathbf{H^i}$ $i = 1, ..., P$ have been computed, samples are assigned to clusters based on the $\mathbf{W}$ matrix; Each sample is associated with the cluster in which it has the highest weight. The algorithm is implemented into the CRAN R package *intNMF* (https://cran-r-project.org/web/packages/IntNMF/index.html).

*Joint and individual variation explained (JIVE)*. JIVE[18] is an extension of PCA to multiomics data. JIVE decomposes each omics matrix into three structures: a joint factor matrix ($\mathbf{J}$), a omics-specific factor matrix ($\mathbf{A}$) and a residual noise ($\mathbf{E}$):

$$\mathbf{X^i} = \mathbf{J} + \mathbf{A^i} + \mathbf{E^i} = \mathbf{U^iS} + \mathbf{A^i} + \mathbf{E^i}, \text{ for } i = 1, ..., P \tag{2}$$

with $\mathbf{E^i}$, $\mathbf{A^i}$ and $\mathbf{U^i}$ are ($n_i \times k$) matrices and $\mathbf{S}$ is a common score matrix explaining variability across multiple data types.
The algorithm minimizes $\|\mathbf{E}\|^2$, with $\mathbf{E^i} = \mathbf{X^i} - \mathbf{U^iS} - \mathbf{A^i}$ and $\mathbf{E} = [\mathbf{E^1}... \mathbf{E^P}]^T$.
JIVE is implemented into the R package *r.jive* (https://cran-r-project.org/web/packages/r.jive/index.html).

*Multiple co-inertia analysis (MCIA)*. MCIA[19], is an extension of co-inertia analysis (CIA) to more than two omics datasets. MCIA factorizes each omics into omics-specific factors

$$\mathbf{X^i} = \mathbf{A^iF^i} + \mathbf{E^i}, \text{ for } i = 1, ..., P, \tag{3}$$

by first applying a dimensionality reduction approach, such as PCA, to each omics matrix $X^i$ separately and then maximizing their co-inertia, i.e., the sum of the squared covariance between scores of each factor:

$$\text{argmax}_{q_1^1...q_P^1} \sum_{k=1}^{P} \text{cov}^2 \left( X_k^i q_k^i, \mathbf{X^i q^i} \right) \tag{4}$$

with $\text{var}(X^i q^i) = 1$ and $q^i$ correspond to the global PCA projections. MCIA is implemented in the R package *omicade4* (https://bioconductor.org/packages/release/bioc/html/omicade4.html).

*Regularized generalized canonical correlation analysis (RGCCA)*. RGCCA[21] is one of the most widely used generalizations of CCA to multiomics data. Similarly to MCIA, RGCCA factorizes each omics into omics-specific factors:

$$\mathbf{X^i} = \mathbf{A^iF^i} + \mathbf{E^i}, \text{ for } i = 1, ..., P. \tag{5}$$

RGCCA maximizes the correlation between the omics-specific factors by finding projection vectors $u^i$ such that the projected data have maximal correlation:

$$\text{argmax}_{i,j} \text{Corr} \left( \mathbf{X^i u^i}, \mathbf{X^j u^j} \right) \text{ for all possible couples} i, j = 1, ..., P. \tag{6}$$

Solving this optimization problem requires inversion of the covariance matrix. However, omics data usually have a higher number of features than samples, and these matrices are therefore not invertible. RGCCA thus apply regularization to CCA. RGCCA is implemented into the CRAN package *RGCCA* (https://cran.r-project.org/web/packages/RGCCA/index.html).

*iCluster*. iCluster[16] decomposes each omics into the product of a factor matrix that is shared across all omics, and omics-specific weight matrices:

$$\mathbf{X^i} = \mathbf{A^iF} + \mathbf{E^i}, \text{ for } i = 1, ..., P. \tag{7}$$

iCluster solves this equation by first deriving a likelihood-based formulation of the same equation and then applying Expectation-Maximization (EM). The method assumes that both the error $E^i$ and the factor matrix $F$ are normally distributed. Finally, clusters are obtained from the factor matrix by applying K-means. The algorithm is implemented into the CRAN package *iCluster* (https://rdrr.io/bioc/iClusterPlus/man/iCluster.html).

*Multiomics factor analysis (MOFA)*. MOFA[15] decomposes each omics into the product of a factor matrix and omics-specific weight matrices:

$$\mathbf{X^i} = \mathbf{A^iF} + \mathbf{E^i}, \text{ for } i = 1, ..., P. \tag{8}$$

MOFA first formulates the equation above in a probabilistic Bayesian model, placing prior distributions on all unobserved variables $A^i$, $F$ and $E^i$. While the factor matrix $F$ is shared across all omics, the sparsity priors in the $A^i$ ensure that both omics-specific and shared factors are retrieved. MOFA solves the probabilistic Bayesian model by maximizing the Evidence Lower Bound (ELBO), which is equal to the sum of the model evidence and the negative Kullback–Leibler divergence between the true posterior and the variational distribution. Despite having a factor matrix $F$ shared among all omics, the sparsity priors in the weights ensure that MOFA will detect both omic-speciifc and shared factors. MOFA is an extension of Factor Analysis to multiomics data, but it is also partially related to iCluster. However, differently from iCluster, MOFA does not assume a normal distribution for the errors but supports combinations of different omics-specific error distributions. The code to run MOFA is available at https://github.com/bioFAM/MOFA. The MOFA package further implements an automatic downstream analysis pipeline for the interpretation of the obtained factor and weight matrices through pathways, top-contributing features or percentage of variance-explained interpretation.

*Tensorial independent component analysis (tICA)*. A natural extension of DR methods to multiomic data is based on the use of tensors, i.e., higher-order matrices. Indeed, all the methods designed for single-omics can be naturally extended to multiomics with tensors. However, this requires to work with omics data sharing both the same samples and features. Here, to overcome this limitation we ran the tensorial algorithm on the correlation-of-correlation matrix, i.e., the matrix having on rows and columns the samples that are common to all the omics data and having in position (i,j) the correlation of sample i with sample j.
We chose tensorial ICA (tICA)[23] to represent the tensor-based methods in our benchmark. Considering the multiomics data organized into a tensor $\mathbf{X}$, the equation solved by tICA is:

$$\mathbf{X} = \mathbf{S} \odot_{i=1}^{P} \Omega_i, \tag{9}$$

where $\mathbf{S}$ is a tensor, with the same dimension of $\mathbf{X}$, and composed of $S_1...S_P$

random variables mutually statistically independent and satisfying $E[S_1...S_P] = 0$ and $Var[S_1...S_P] = I$ and $\odot$ denotes the tensor contraction operator.

Thus, tICA searches for independent signals. Since biological processes are generally non-Gaussian and often sparse, the assumption of tICA can improve the deconvolution of complex mixtures and hence better identify biological functions and pathways underlying the multiomics data. Given that multiple tensorial versions of ICA exist, we considered the tensorial fourth-order blind identification (tFOBI), whose implementation in R was reported by Teschendorff et al.[23].

*Multi-study factor analysis (MSFA).* MSFA[20] is a generalization of Factor Analysis (FA), which models the omics matrices $X^i$ as the sum of data-specific and shared factors:

$$\mathbf{X}^i = \Phi \mathbf{F}^i + \Lambda^i \mathbf{L}^i + \mathbf{E}^i, \text{ for } i = 1, ..., P. \qquad (10)$$

where $E^i$ has a multivariate normal distribution and the marginal distributions of $F^i$, $L^i$, and $X^i$ are multivariate normal. MSFA is implemented in R and available at https://github.com/rdevito/MSFA.

*Data fusion (scikit-fusion).* The data-fusion approach (scikit-fusion)[22] is based on two steps. First, two groups of matrices are constructed from the multiomics data: relation ($R$) and constraint ($C$) matrices. The $R$ matrix encodes relations inferred between features of different omics (e.g., genes to proteins), while the matrix $C$ describes relations between features of the same omics (e.g., protein–protein interactions). The matrix $C$ thus corresponds to the side information considered by scikit-fusion in the factorization. Then, tri-matrix factorization is used to simultaneously factorize the various relation matrices $R$ under constraints $C$. Given that the $R$ and $C$ matrices are block-matrices, with element $R_{ij}$ containing a relation between the elements of the i-th omics and those of the j-th, the matrix-tri-factorization is applied separately to each block:

$$R_{ij} \approx \mathbf{G}_i S_{ij} \mathbf{G}_j, \qquad (11)$$

with $G_i$ shared across all the $R_{ip}$ for $p = 1...P$ (matrices that relate the i-th object to others).

Hence, scikit-fusion can naturally combine additional side information in the factorization of the multiomics data, such as protein–protein interactions, Gene Ontology annotations. It is implemented in Python and available at https://github.com/marinkaz/scikit-fusion.

**Factor selection for performance comparisons.** The jDR approaches make different assumptions on the cross-omics constraints of the factors. The various jDR can be thus classified in shared factors, omics-specific factors and mixed factors approach. To use the factor matrices to compare the performances of the various jDR methods, e.g., to cluster the samples based on the factors, we had to select which factor matrix to use for each jDR. Shared factors jDR methods compute a unique factor matrix, which is used in our benchmark. Omics-specific jDR methods compute a factor matrix for each omics dataset. In these cases, we selected the factor matrix associated with transcriptomic data for our benchmark. However, jDRs with omics-specific factors maximize correlation or co-inertia between the various omics-specific factor matrices. The values of the transcriptomic factor matrix are then influenced by the other omics. Finally, for mixed factors jDRs methods, we considered the joint factor matrix $F$. As a consequence, all jDR methods with omics-specific and mixed factors contain more information in their factorization than that considered here for sake of comparison.

**Data simulation.** The simulated multiomics datasets have been produced by the *InterSIM* CRAN package[24]. *InterSIM* simulates multiple interrelated data types with realistic intra- and inter-relationships based on the DNA methylation, mRNA gene expression, and protein expression from TCGA ovarian cancer data. We generated 100 simulated datasets, with a number of clusters set by the user. We considered five, ten, and fifteen clusters in this study. The proportion of samples belonging to each subtype is also set by the user, while we considered here two conditions with equally sized clusters and variable random sizes, respectively.

**Clustering of factor matrix.** To identify the clusters of samples starting from the jDR factor matrix, we applied k-means clustering to the factor matrix (*kmeans* function in R). We chose k-means for clustering in agreement with the use of k-means in iCluster and euclidean distance in intNMF for clustering. As k-means clustering is stochastic, we performed clustering 1000 times and computed a consensus consisting in the most frequent associations between samples and clusters

**Comparing jDR algorithm clusters to ground-truth clusters.** The matching between the ground-truth clustering and the clustering inferred by the various jDR algorithms is measured with the Jaccard Index (JI) and Adjusted Rand Index (ARI).

JI is a similarity coefficient between two finite sets A and B, defined by the size of the intersection of the sets, divided by the size of their union: $JI(A,B) = \frac{A \cap B}{A \cup B}$. It takes its values in [0;1].

Given two partitions/clustering results $(z, z')$ obtained on the same data of $n$ total elements, the Rand Index (RI) is defined as:

$$R(z, z') = \frac{a + b}{\binom{n}{2}},$$

where $\frac{n}{2}$ is the binomial coefficient measuring the number of unordered pairs in a set of $n$ elements, $a$ refers to the number of times a pair of elements belongs to a same cluster across the two partitions/clustering results $(z, z')$ and $b$ refers to the number of times a pair of elements belongs to different clusters across the two partitions/clustering results $(z, z')$.

The RI is always comprised between 0 and 1, with 1 corresponding to a perfect matching between two partitions/clustering results. However, for random partitions, the expected value of the RI is close to 1. To overcome this drawback, the Adjusted Rand Index (ARI) has been proposed. The ARI is the normalized difference between the RI and its expected value, according to the formula:

$$ARI = \frac{RI - \text{expected value } RI}{\max RI - \text{expected value } RI} \qquad (12)$$

**Selection of the clinical annotations.** The clinical annotations selected for benchmark testing are "age of patients", "days to new tumor", "gender" and "neo-adjuvant therapy administration". This set of annotations is obtained after excluding redundant annotations (e.g., "age_at_initial_pathologic_diagnosis" and "years_of_initial_pathologic_diagnosis"), annotations having missing values for more than half of the samples, and annotations having no biological relevance (e.g., "vial_number", "patient_id"). Four clinical annotations are available for nine or ten out of ten cancer types, while the others are only present for six or fewer cancer types (with most of them being available only for one or two cancer types).

**Selectivity score.** We define the selectivity as:

$$S = \frac{Nc + N_f}{2L} \qquad (13)$$

where Nc is the total number of clinical annotations associated with at least a factor, Nf the total number of factors associated with at least a clinical annotation, and L the total number of associations between clinical annotations and factors. S has a maximum value of 1 when each factor is associated with one and only one clinical/biological annotation, and a minimum of 0 in the opposite case. An optimal method should thus maximize its number of factors associated with clinical/biological annotations without having a too low selectivity.

**Testing the biological enrichment of metagenes.** To test if metagenes are enriched in biological annotations, we used *PrerankedGSEA*, implemented in the *fgsea* R package. In *preranked GSEA*, each metagene is considered as a ranking of genes, and the significance of the association of a biological annotation with the higher or lower part of the ranking is tested. We considered as biological annotations Reactome pathways, Gene Ontology (GO), and cancer Hallmarks, all obtained from MsigDB[35].

**Quality of single-cell clusters.** To evaluate the quality of the 2D data distribution obtained from single-cell multiomics data, we employed the C-index measure[36] an internal clustering evaluation index comparing the distance between intracluster points and interclusters points. It takes its values in [0,1] and should be minimum in an optimal clustering. Of note, to compute the C-index we do not need to perform a clustering on the jDR factor matrix, but only to compute the distance between cells known to belong to the same or different cell lines.

Instead, in the computation of the ARI, we had first to define a clustering based on the jDR output, which was performed with k-means, and then compared to the cell lines of origin.

**Reporting summary.** Further information on research design is available in the Nature Research Reporting Summary linked to this article.

# Data availability

The simulated data are produced using the R package InterSIM V2.2 and can be reproduced using our jupyter notebook (https://github.com/ComputationalSystemsBiology/momix-notebook)[37]. The cancer TCGA data were dowloaded from http://acgt.cs.tau.ac.il/multi_omic_benchmark/download.html. The single-cell data are available in the data/ folder of our github repository (https://github.com/ComputationalSystemsBiology/momix-notebook)[37]. Annotations (i.e., Gene Ontology, Reactome and MSigDB Hallmarks) were fetched from fgsea V1.16. The remaining data are available in the article, Supplementary Information and Jupyter notebook[37].

# Code availability

All the analyses can be reproduced with the momix Jupyter notebook available at https://github.com/ComputationalSystemsBiology/momix-notebook[37]. The same github link

also contains the instructions to install the momix conda environment available on Anaconda cloud (https://anaconda.org/lcantini/momix). With this conda environment, the user will get all the required R and Python packages automatically installed, in the same version that we used for this benchmark. The packages used are: R packages (iCluster V2.1, intNMF V1.2, r.jive V2.1, omicade4 V1.3, MOFA V1, MSFA V1, RGCCA2.1.2, InterSIM V2.2, fgsea V1.16). Python packages (scikit-fusion V1).

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

## Acknowledgements

The project leading to this publication has received funding from the Agence Nationale de la Recherche (ANR) project scMOmix and the Excellence Initiative of Aix-Marseille University—A*Midex, a French "Investissements d'Avenir" programme.

## Author contributions

A.B-., E.R., and L.C. designed, planned, and wrote the manuscript. D.T. and P.Z. revised the manuscript. L.C. and P.Z. conducted data analysis. A.N. and C.H. participated in the development. All authors read and approved the final manuscript.

## Competing interests

The authors declare no competing interests.
