## [Peer Review File · Nature Communications]

Editorial Note: Reviewer #3 was recruited in the third round of review to adjudicate the response of the authors to Reviewer #1

Reviewers' comments:

Reviewer #1 (Remarks to the Author):

This work compares different dimensionality reduction algorithms tailored from multi-omics data using different criteria and datasets. Dimensionality reduction algorithms are a set of methods whose purpose is to learn or infer a low-dimensional representation of some high-dimensional data while containing as much information possible. These algorithms can be formulated using different mathematical principles and thus have different advantages or disadvantages against each other. Studying these comparisons is an important practical task as it will guide bioinformaticians in the choice of what algorithm to use. Furthermore, these benchmarks are useful for researchers to compare new algorithms as they are developed. However, there is no conceptual advance in this exercise, which would make this paper better suited for a specialist journal. Furthermore, some of the tests they have designed are not appropriate as detailed below, and therefore, this study does not represent a thorough benchmark.

Major technical limitations

This work has major limitations in three crucial aspects. First, the authors chose a set of algorithms that seem to use linear transformations of the input data. In the last years, the field of machine learning has shown that algorithms with non-linear transformations in many cases excel at classification, regression and un-supervised learning tasks (e.g. random forests, support vector machines, the family of deep learning algorithms, t-SNE, etc.). The paper would benefit from comparing the current set of algorithms with at least one non-linear dimensionality reduction algorithms even if they are not specifically tailored for multi-omics. To enable this comparison the authors simply could concatenate the multi-omics data into a large matrix. The algorithms suggested for comparisons include t-distributed stochastic neighbour embedding (t-SNE, van der Maaten and Hinton 2008) and, if possible, either restricted Boltzmann machine (RBM, Carreira-Perpinan and Hinton 2005) or a shallow variational autoencoder (VAE, Kingma and Welling 2014). These algorithms are common use in machine learning and have many implementations in different programming languages available.

The second and more important limitation is that not all the experiments test the dimensionality reduction capacity of different algorithms. The first and last experiments for testing different algorithm on simulated datasets are appropriate, as they require each algorithm to match the inferred clusters against the ground truth clusters. Such approaches have proven useful before to test dimensionality reduction algorithms (van der Maaten and Hinton 2008). However, the rest of the experiments does not test the effectiveness of the dimensionality reduction algorithms. The authors tested if clustering performed in the resulting reduced datasets would match what they consider as ground truth clusters in the breast cancer TCGA dataset, which they assume to be receptor status subtype. This is not necessarily the case. Dimensionality reduction is meant to represent the multi-omics data and not the underlying biological processes. This might sound incorrect from a biological perspective as multi-omics is often used to produce patterns which correspond to biological processes. Yet mathematically, the aim of dimensionality reduction algorithms is to reproduce the data itself in an unsupervised manner. The correct comparison would be simply to see if the clusters obtained from the latent factors match clusters obtained from the data. This is a fundamental theoretical error that questions the validity of this work as a thorough benchmarking exercise. This error is repeated in the subsequent experiments, where researchers

aim to match the low-dimensional outputs with survival, clinical variables or biological processes. These results can be interesting for the readers, but formally they do not test the purpose of the dimensionality reduction algorithms. It is inappropriate to take these experiments as a reflection of the effectiveness of each algorithm.

Finally, and importantly the authors focused only on clustering which is not the prime application for dimensionality reduction algorithms. Most clustering algorithms can scale very well to large dimensional datasets, and we do not need dimensionality reduction to perform clustering. Perhaps the most important and commonly used application of dimensionality reduction is data visualization. PCA and t-SNE have been used for years to visualize otherwise impossible to represent patterns. To perform a thorough benchmark the authors, need to compare or at least show this aspect.

On a minor note, "neo-adjuvant therapy somministration" is a phrase unknown to Pubmed, and even in Google the only hit is the bioRxiv deposition of this paper.

For these reasons, the paper needs major revisions. In other to see how these algorithms compare to the current machine learning literature, the authors need to include dimensionality reductions algorithms that are non-linear, specially t-SNE and if possible RBM or VAE. Furthermore, the authors need to re-do some of the experiments to make for appropriate comparisons between algorithms. The experiments that match latent factors with clinical or biological data are interesting and can be used to support the results but should not be used to test dimensionality reduction efficacy. Finally, the authors need to include at least one comparison between the data visualization capabilities of these algorithms.

Reviewer #2 (Remarks to the Author):

In this manuscript Cantini et al benchmark several methods for multi-omic dimensionality reduction with application to simulated and real cancer data and also to single cell data.

The manuscript is in principle interesting; however, I feel before publication it is important to address several points:

1) I think the main figures should be significantly improved.

For example, in figure 1 you should present the general workflow of your benchmark strategy highlighting the three main objectives illustrated in the abstract:

- Benchmark on simulated data
- Analysis on TCGA cancer data to assess the ability of methods to predict survival, clinical annotations and known pathways/biological processes.
- Benchmark on single-cell data analysis

Also, from the figure it is not clear if the different datasets are collected simultaneously for each individual.

In figure 2 it may be nice to illustrate how the simulation is generating the data, the clustering

approach adopted and not just a scatter plot with the results.

Same remarks apply for the other main figures presented.

2) “the genuine joint analysis of multi-omics data remains very rare” define genuine in this context

3) I think this is an overstatement: “Thus, DR simultaneously provides information on all the key objectives mentioned above, namely the classification of samples into subtypes, their association with outcome/survival, as well as the reconstruction of their underlying molecular mechanisms.”

Please explain what you mean by information, after DR you need to perform downstream analyses to recover sub-types, have information on survival or molecular mechanisms. Maybe you can just say that this is a common step before common downstream analyses?

4) Lines 129-130 I am confused by the proposed formulation. Usually different assays will give you different number of features. Here it is important to discuss how the different methods are dealing with different number of features across assays. This is a key concept and needs to be discussed in depth.

This is partially explained later in lines 175-178, but this should be expanded and moved before.

5) Lines 132-133 Briefly explain the concept of factor and their interpretation for non-technical readers.

6) Lines 192-198, explain briefly the core idea of the simulation package used. Also, what omic assays were simulated for each sample? Discuss the limitations of this simulation approach. Also it is not clear how a simulation strategy can “avoid making assumptions on the distribution of the simulated data”

7) Lines 206, not clear if the choice of clustering approach has a strong influence on the results. Please include other common clustering approaches, hierarchical (with average/ward linkage) and a simple k-means.

8) To evaluate clustering performance, more robust metrics have been proposed. Please use the Adjusted Rand Index.

9) Line 220: For practical purposes, results with side information should be provided to show if including additional information improve the results, given that this approach can leverage this additional information.

10) Lines 232-233. What about using the subtypes defined by papers published by the TCGA consortium?

11) Lines 241-244 The low performance of all the methods, suggest that the simulation doesn't recapitulate well real case scenarios.

12) It would be nice to show for the factors recovered and associated with survival/clinical annotations/biological processes and pathways what are the genes (or other features) associated and discuss if they have any biological relevance. Also, it is necessary to discuss how to recover important features based on these factors in each of the omic datasets used before the integration.

13) Lines 322-324 In addition,... Explain the rationale of this statement and cite relevant work that support this claim.

14) The section 4 needs to be significantly expanded. It doesn't provide in the current form any useful information. Current experimental multi-omics approaches and computational methods for their analysis currently used in the single cell field are marginally described. Please, take a look at this paper that systematically dissects this point:

[https://www.cell.com/trends/biotechnology/fulltext/S0167-7799\(20\)30057-3?rss=yes](https://www.cell.com/trends/biotechnology/fulltext/S0167-7799(20)30057-3?rss=yes)

15) Line 337, why do you use the C-index? Before you were using the Jaccard coefficient. I would suggest using as I proposed before the Adjusted Rand Index.

16) Also approaches specifically developed for single cell data are not tested, e.g. Conos, Seurat, Liger and MOFA+ should be tested. See again the paper mentioned in 14).

17) Line 418-420 for the methods presented it is necessary to explore what was recovered in other omics-specific factor and explain to the reader how to potentially interpret this additional information.

A key point of this paper is to show how different methods perform for data integration. However, how the transcriptome associated factors would change excluding additional omics datasets? Is it true that the performance of these integrative approaches truly benefits of the data integration? This should be shown and discussed. You should have as a baseline DR methods based only on transcriptomic data e.g. NMF, ICA or PCA.

18) Line 430, please explain "slightly non linear signal", which dataset are you referring to?

Reviewer #1 (Remarks to the Author):

This work compares different dimensionality reduction algorithms tailored from multi-omics data using different criteria and datasets. Dimensionality reduction algorithms are a set of methods whose purpose is to learn or infer a low-dimensional representation of some high-dimensional data while containing as much information possible. These algorithms can be formulated using different mathematical principles and thus have different advantages or disadvantages against each other. Studying these comparisons is an important practical task as it will guide bioinformaticians in the choice of what algorithm to use. Furthermore, these benchmarks are useful for researchers to compare new algorithms as they are developed. However, there is no conceptual advance in this exercise, which would make this paper better suited for a specialist journal. Furthermore, some of the tests they have designed are not appropriate as detailed below, and therefore, this study does not represent a thorough benchmark.

We thank the reviewer for highlighting the importance of benchmarking studies, but also assert that our work does propose conceptual advances as well as appropriated tests. After analyzing the comments from reviewer 1 in detail, we interpret the major concerns of the reviewer as arising from a confusion between the more established use of dimensionality reduction applied to single omics for data visualization and the use of dimensionality reduction for multi-omics integration.

On one side, dimensionality reduction approaches are used for omics data visualization (i.e., using approaches such as PCA, ICA, NMF, tSNE, or UMAP) [1]. However, the benchmark proposed here concerns another purpose and thus a different set of tools, dedicated to the dimensionality reduction for multiple omics data integration. Besides compressing omics data to a lower dimension, dimensionality reduction approaches are indeed increasingly used to extract factors of biological interest, e.g. factors associated with specific biological processes or with prognosis/drug response. This growing interest is demonstrated by a large number of high-impact publications using dimensionality reduction to extract factors and characterize them from a prognosis/clinical/biological point of view [2-6]. For instance, in [3], Argelauguet R. and colleagues apply their tool (MOFA) to Chronic Lymphocytic Leukaemia multi-omics datasets and search for associations with prognosis and clinical annotations (see Figure 4 in [3]).

To avoid any confusion, we are now clarifying the rationale for using jDR in multi-omics data integration and the relevance of the factors identified by jDR in the Results section of the manuscript (pages 3-4):

“The rationale behind the use of jDR in biology is that the state of a biological sample is determined by multiple concurrent biological factors, from generic processes (e.g., proliferation and inflammation) to cell-specific processes. When measuring omics data, we take a snapshot of the state of a biological sample and thus detect a convoluted mixture of various biological factors active in the sample. The goal of jDR is to deconvolute this mixture and expose the different biological factors contributing to the state of the biological sample.”

In addition, we further precise in the introduction:

“Of note, DR is employed in computational biology in different contexts, such as data visualization or matrix completion. We focus here on DR for multi-omics data integration.”

References:

1. Sun, Shiquan, et al. "Accuracy, robustness and scalability of dimensionality reduction methods for single-cell RNA-seq analysis." *Genome Biology* 20.1 (2019): 269.
2. Stein-O'Brien, Genevieve L., et al. "Enter the matrix: factorization uncovers knowledge from omics." *Trends in Genetics* 34.10 (2018): 790-805.
3. Argelaguet, Ricard, et al. "Multi-Omics Factor Analysis—a framework for unsupervised integration of multi-omics data sets." *Molecular systems biology* 14.6 (2018).
4. Welch, Joshua D., et al. "Single-cell multi-omic integration compares and contrasts features of brain cell identity." *Cell* 177.7 (2019): 1873-1887.
5. Cantini, Laura, et al. "Assessing reproducibility of matrix factorization methods in independent transcriptomes." *Bioinformatics* 35.21 (2019): 4307-4313.
6. Argelaguet, Ricard, et al. "Multi-omics profiling of mouse gastrulation at single-cell resolution." *Nature* (2019): 1-5.

Major technical limitations

This work has major limitations in three crucial aspects. First, the authors chose a set of algorithms that seem to use linear transformations of the input data. In the last years, the field of machine learning has shown that algorithms with non-linear transformations in many cases excel at classification, regression and un-supervised learning tasks (e.g. random forests, support vector machines, the family of deep learning algorithms, t-SNE, etc.). The paper would benefit from comparing the current set of algorithms with at least one non-linear dimensionality reduction algorithms even if they are not specifically tailored for multi-omics. To enable this comparison the authors simply could concatenate the multi-omics data into a large matrix. The algorithms suggested for comparisons include t-distributed stochastic neighbour embedding (t-SNE, van der Maaten and Hinton 2008) and, if possible, either restricted Boltzmann machine (RBM, Carreira-Perpinan and Hinton 2005) or a shallow variational autoencoder (VAE, Kingma and Welling 2014). These algorithms are common use in machine learning and have many implementations in different programming languages available.

We do think that this comment arises from the confusion with the use of dimensionality reduction (DR) approaches for data visualisation. Indeed, the reviewer is proposing tSNE as a good example of non-linear method to compare in our benchmark, while tSNE is a visualization tool (in the abstract of the tSNE paper, the authors state clearly “We present a new technique called "tSNE" that visualizes high-dimensional data by giving each datapoint a location in a two or three-dimensional map”). Concerning VAE or RBM, we studied the articles suggested by the reviewer, but we found only applications to digits and image reconstruction. We checked for other references, and found that the VAE methods are mentioned in a recent

review of linear and non-linear multi-omics data integration methods [1], but here also referring to their applications to images and digits [2-3]. Other recent applications are related to single-cell analyses [4,5], but, overall, we did not find any application to multi-omics integration. Of note, extending an algorithm designed for single omics to multi-omics is not a trivial task. In addition, The other suggested methods (e.g. decision trees (DT), random forests) are classification methods and thus do not provide joint information on features and samples, contrarily to the jDR method considered in our work [6]. Finally, to the date, as stated in the discussion of the manuscript, no multi-omics data integration method is able to handle non-linearity.

References:

1. Mirza, Bilal, et al. "Machine learning and integrative analysis of biomedical big data." *Genes* 10.2 (2019): 87.
2. Hinton, Geoffrey E., and Ruslan R. Salakhutdinov. "Reducing the dimensionality of data with neural networks." *science* 313.5786 (2006): 504-507
3. Wang, Yasi, Hongxun Yao, and Sicheng Zhao. "Auto-encoder based dimensionality reduction." *Neurocomputing* 184 (2016): 232-242.
4. Eraslan, Gökçen, et al. "Single-cell RNA-seq denoising using a deep count autoencoder." *Nature communications* 10.1 (2019): 1-14.
5. Grønbech, Christopher H., et al. "scVAE: Variational auto-encoders for single-cell gene expression datas." *bioRxiv* (2018): 318295.
6. Liaw, Andy, and Matthew Wiener. "Classification and regression by randomForest." *R news* 2.3 (2002): 18-22.

The second and more important limitation is that not all the experiments test the dimensionality reduction capacity of different algorithms. The first and last experiments for testing different algorithm on simulated datasets are appropriate, as they require each algorithm to match the inferred clusters against the ground truth clusters. Such approaches have proven useful before to test dimensionality reduction algorithms (van der Maaten and Hinton 2008).

However, the rest of the experiments does not test the effectiveness of the dimensionality reduction algorithms. The authors tested if clustering performed in the resulting reduced datasets would match what they consider as ground truth clusters in the breast cancer TCGA dataset, which they assume to be receptor status subtype. This is not necessarily the case. Dimensionality reduction is meant to represent the multi-omics data and not the underlying biological processes. This might sound incorrect from a biological perspective as multi-omics is often used to produce patterns which correspond to biological processes. Yet mathematically, the aim of dimensionality reduction algorithms is to reproduce the data itself in an unsupervised manner. The correct comparison would be simply to see if the clusters obtained from the latent factors match clusters obtained from the data. This is a fundamental theoretical error that questions the validity of this work as a thorough benchmarking exercise. This error is repeated in the subsequent experiments, where researchers aim to match the low-dimensional outputs with survival, clinical variables or biological processes. These results can be interesting for the readers, but formally they do not test the purpose of the dimensionality reduction algorithms. It is inappropriate to take these experiments as a reflection of the effectiveness of each algorithm.

As discussed in the first section of the Results in our manuscript, the aim of dimensionality reduction (DR) algorithms is to compress omics data in an unsupervised manner without losing

their biological information. The output of DR can then be used for visualization, as mentioned by the Reviewer ("Dimensionality reduction is meant to represent the multi-omics data and not the underlying biological processes"). However, DR can be used, and has been extensively used to extract factors of biological interest, e.g. associated with specific biological processes or with prognosis/drug response. Our tests on the association of DR factors with survival, clinical annotations, biological annotations, and cancer subtypes, are thus relevant in the context of the application of DR to multi-omics data integration. Different examples highlighting the use of DR approaches on single omics (mainly transcriptome) data analyses, and more recently of multi-omics data integration, in this context are detailed below:

[1] Biton A. et al. used ICA, a dimensionality reduction approach, to characterize the biological processes, pathways, genomics alterations and subtypes of bladder cancers. Their analysis of the ICA factors also identified the prognostic role of the Transcription Factor PPARG in luminal tumors, which was experimentally validated.

[2] Cantini L. et al. compared the reproducibility of various DR methods (ICA, NMF, PCA) also based on their ability to retrieve biological signals, such as proliferation and stromal infiltration. In colorectal cancer (CRC), these factors were further found to highlight the biological processes underlying CRC and its subtypes. These results are in agreement with what previously found in the CRC subtypes papers.

[3] Stein-O'Brien, G. L. et al. deeply explain how to use factors extracted from DR to interpret their association with survival, subtypes or biological annotations (see Figure 2 of the paper).

[4] Argelaguet R. et al. use MOFA to integrate multi-omics bulk data from CLL and they characterize the association of the identified factors with survival, drug response and biological processes. Of note, MOFA automated all this downstream analysis for the annotation of the factors as part of its tool. The wide use of MOFA (119 citations form 2018), thus further highlights how much DR tools are used to interpret the biological information encoded in the factors and not just to reduce the dimension of the data.

[5] Argelaguet R. et al. present an example of the importance and relevance of the biology associated with DR factors in the context of single-cell multi-omics integration. The authors applied MOFA to single-cell multi-omics data from mouse embryonic gastrulation and, analysing the biology of the factors extracted by MOFA, they identified six factors capturing: the emergence of the three germ layers, anterior–posterior axial patterning, notochord formation, mesoderm patterning and cell cycle. Using the projections on epigenetic data they could then further single-out epigenetic markers regulating such processes.

References:

[1] Biton, Anne, et al. "Independent component analysis uncovers the landscape of the bladder tumor transcriptome and reveals insights into luminal and basal subtypes." *Cell reports* 9.4 (2014): 1235-1245.

[2] Cantini, Laura, et al. "Assessing reproducibility of matrix factorization methods in independent transcriptomes." *Bioinformatics* 35.21 (2019): 4307-4313.

[3] Stein-O'Brien, Genevieve L., et al. "Enter the matrix: factorization uncovers knowledge from omics." *Trends in Genetics* 34.10 (2018): 790-805.

[4] Argelaguet, Ricard, et al. "Multi-Omics Factor Analysis—a framework for unsupervised integration of multi-omics data sets." *Molecular systems biology* 14.6 (2018): e8124.

[5] Argelaguet, Ricard, et al. "Multi-omics profiling of mouse gastrulation at single-cell resolution." *Nature* (2019): 1-5.

Finally, and importantly the authors focused only on clustering which is not the prime application for dimensionality reduction algorithms. Most clustering algorithms can scale very well to large dimensional datasets, and we do not need dimensionality reduction to perform clustering. Perhaps the most important and commonly used application of dimensionality reduction is data visualization. PCA and t-SNE have been used for years to visualize otherwise impossible to represent patterns. To perform a thorough benchmark the authors, need to compare or at least show this aspect.

As stated previously, we disagree with the reviewer on this point. The reviewer confused two independent purposes of dimensionality reduction in computational biology. Dimensionality reduction for single omics (PCA, NMF, ICA) is frequently used to reduce the dimension of data before visualization. Such methods are currently undergoing a great development, especially in relation with single-cell data [1]. tSNE and UMAP are two of the most popular tools used for this purpose. However, dimensionality reduction is also used to jointly extract biological factors from multi-omics data, which is the focus of our paper. In this context, comparing the joint DR tools for multi-omics integration based on their performance for data visualization is irrelevant. In support of this view, the authors of the articles presenting the tools benchmarked in our manuscript don't present any example of application of their tools for data visualization, and they don't compare performances with existing alternative tools based on their visualization abilities.

References

[1] Sun, Shiquan, et al. "Accuracy, robustness and scalability of dimensionality reduction methods for single-cell RNA-seq analysis." *Genome Biology* 20.1 (2019): 269.

On a minor note, "neo-adjuvant therapy somministration" is a phrase unknown to Pubmed, and even in Google the only hit is the bioRxiv deposition of this paper.

We used the annotation "neo-adjuvant therapy somministration" to refer to the TCGA standard annotation "neo-adjuvant therapy history". We now changed it to "neo-adjuvant therapy administration" for simplification.

For these reasons, the paper needs major revisions. In other to see how these algorithms compare to the current machine learning literature, the authors need to include dimensionality reductions algorithms that are non-linear, specially t-SNE and if possible RBM or VAE. Furthermore, the authors need to re-do some of the experiments to make for appropriate comparisons between algorithms. The experiments that match latent factors with clinical or biological data are interesting and can be used to support the results but should not be used

to test dimensionality reduction efficacy. Finally, the authors need to include at least one comparison between the data visualization capabilities of these algorithms.

As we detailed in our point-by-point responses, all these concerns likely arise from a confusion between two different purposes of DR methods. This said, the confusion might arise from a lack of clarity in our introduction, which has been addressed in this revised version of our manuscript.

“Of note, DR is employed in computational biology in different contexts, such as data visualization or matrix completion. We focus here on DR for multi-omics data integration.”

Reviewer #2 (Remarks to the Author):

In this manuscript Cantini et al benchmark several methods for multi-omic dimensionality reduction with application to simulated and real cancer data and also to single cell data.

We thank the reviewer for his/her detailed review of our work and for his/her pertinent suggestions. We provide below a point-by-point to all the concerns and propose revised versions of the manuscript (with changes highlighted in blue), supplementary material and notebook. Overall, following the reviewer's comments:

- We drastically improved the Figures of the paper, adding four new Figures or Panels and three new Supplementary Figures. More precisely, following the Reviewer's suggestions, we added three Figures/Panels to illustrate the protocol and experiments we performed. The remaining new Figures/SuppFigures present results from additional experiments.
- We performed 12 new experiments. In particular,
 - (i) we computed the Adjusted Rand Index (ARI) for the evaluation of all our clustering results;
 - (ii) we applied PCA and compared its performances with jDR methods in prediction of survival;
 - (iii) we tested the association of the jDR factors with the TCGA integrative breast cancer subtypes;
 - (iv) following the Reviewer's suggestion, we extended our benchmark to include two single-cell integrative state-of-the-art methods, LIGER and Seurat, and compared their clustering performances with those of the jDR methods. We found that jDR methods designed for bulk data perform comparably and in some cases better than methods devoted to single-cell data, overall emphasizing the novelty and interest of our analysis.

The manuscript is in principle interesting; however, I feel before publication it is important to address several points:

1) I think the main figures should be significantly improved.

For example, in figure 1 you should present the general workflow of your benchmark strategy highlighting the three main objectives illustrated in the abstract:

- Benchmark on simulated data
- Analysis on TCGA cancer data to assess the ability of methods to predict survival, clinical annotations and known pathways/biological processes.
- Benchmark on single-cell data analysis

Also, from the figure it is not clear if the different datasets are collected simultaneously for each individual.

In figure 2 it may be nice to illustrate how the simulation is generating the data, the clustering approach adopted and not just a scatter plot with the results.

Same remarks apply for the other main figures presented.

Following the suggestions of the Reviewer, we have substantially improved all the figures present in the paper.

For Figure 1, we divided the figure into two panels. Figure 1A contains the previous Figure 1, in which we modified the left side to more clearly show that the datasets are collected from the same samples (we also clarified this information in the Figure legend). Figure 1B contains a graphical description of the complete workflow.

B

For Figure 2, we now better describe the simulated datasets generated by InterSIM, how the results of jDR are used to obtain the sample clustering, and how the quality of the clustering is evaluated. We also added the Adjusted Rand Indices (ARI) in parenthesis after the method names.

We added a new Figure, numbered Figure 3, summarizing all the tests performed on TCGA cancer data (association with survival, clinical annotations, biological annotations), together with an illustration to explain the selectivity score.

For the former Figure 5 (now Figure 6), we now further report the Adjusted Rand Indices (ARI) obtained from single-cell data by the various jDR approaches, as well as the results obtained with the new methods LIGER and Seurat.

We finally added a new **Supplementary Figure 1** which contains the comparisons with the two reference breast cancer subtypings. The panel A focuses on the comparisons of the different jDR clustering outputs with the ER/PR/HER-2 subtyping, as previously presented in the Supplementary Figure 1. The panel B focuses on the comparisons with the subtypes defined in the article published by the TCGA consortium, namely the Cluster of Clusters (COCA) [Cancer Genome Atlas Network. Comprehensive molecular portraits of human breast tumours. *Nature* **490**, 61–70 (2012)].

2) “the genuine joint analysis of multi-omics data remains very rare” define genuine in this context

With “genuine joint analysis of multi-omics data”, we meant intermediate integration, i.e. the development of algorithms that can co-analyse different omics at the same time, as opposed to early or late integration algorithms. This point is now clarified in our revised manuscript (page 2):

“While multi-omics data are becoming more accessible, studies combining different omics are more common. This multi-omics integration is frequently performed by sequentially combining

results obtained on single omics (a.k.a. late or early integration), but the genuine joint analysis of multi-omics data (a.k.a. intermediate integration) remains very rare³.

3) I think this is an overstatement: “Thus, DR simultaneously provides information on all the key objectives mentioned above, namely the classification of samples into subtypes, their association with outcome/survival, as well as the reconstruction of their underlying molecular mechanisms.”

Please explain what you mean by information, after DR you need to perform downstream analyses to recover sub-types, have information on survival or molecular mechanisms. Maybe you can just say that this is a common step before common downstream analyses?

We agree with the reviewer that the first sentence could appear overstated, as downstream analyses are necessary. We wanted to refer to the fact that jDR approaches allow us to automatically get both a factor matrix providing sample-level information and weight matrices providing features-level information. Our goal was to emphasize that such interpretation is missing from similarity-based methods, such as those based on networks, that build a similarity matrix between samples and ignore the original features. In order to adjust the statement and clarify this point, we changed it to (page 3):

“Thus, DR combined with dedicated downstream analyses provide information on all the key objectives mentioned above, namely the classification of samples into subtypes, their association with outcome/survival, as well as the reconstruction of their underlying molecular mechanisms.”

4) Lines 129-130 I am confused by the proposed formulation. Usually different assays will give you different number of features. Here it is important to discuss how the different methods are dealing with different number of features across assays. This is a key concept and needs to be discussed in depth.

This is partially explained later in lines 175-178, but this should be expanded and moved before.

Indeed, this is a key point, that we previously only described in the section dedicated to tensor ICA. We are now adding several sentences discussing the different numbers of features in the first section of the results (page 5):

“In addition, the number of features n_i in the various omics is highly variable, going for instance in TCGA from 800 microRNAs to 5000 CpGs to 20.000 genes. Omics containing more features will have a higher weight in the jDR output. To overcome this issue, in the following, we will first select features based on their variability, and thus make the number of features of the various omics comparable.”

We also changed the mathematical formulation to clarify the concepts (page 4):

“We consider P omics matrices X^i , $i = 1, \dots, P$ of dimension $n_i \times m$, with n_i features (e.g. genes, proteins) and m samples.”

5) Lines 132-133 Briefly explain the concept of factor and their interpretation for non-technical readers.”

Done. We now better explain the rationale for using jDR on biological data for non-technical readers (page 3):

“The rationale behind the use of jDR in biology is that the state of a biological sample is determined by multiple concurrent biological factors, from generic processes (e.g., proliferation and inflammation) to cell-specific processes. When measuring omics data, we take a snapshot of the state of a biological sample and thus detect a convoluted mixture of various biological signals active in the sample. The goal of jDR is to deconvolute this mixture and expose the different biological signals contributing to the state of the biological sample.”

We also improved the following explanation (page 4):

“Factors and metagenes represent the projections on the sample space and gene space, respectively, of the biological signals present in the profiled samples.”

6) Lines 192-198, explain briefly the core idea of the simulation package used. Also, what omic assays were simulated for each sample? Discuss the limitations of this simulation approach. Also it is not clear how a simulation strategy can “avoids to making assumptions on the distribution of the simulated data”

InterSIM starts from real omics data (DNA methylation, transcriptome and protein expression) extracted from TCGA ovarian cancer datasets. It then generates clusters and associates features to these clusters by shifting the features' mean values by a fixed amount. Such transformation is performed conserving the covariance matrices between all couples of omics. Therefore, InterSIM generates data with realistic relationships between features in the same omics and across omics. Importantly, InterSIM does not generate synthetic data with a fixed distribution, e.g. Gaussian sample distribution.

We now added explanations of data simulations thanks to the interSIM package in the new version of the Figure 2, and better detail how and which omics data are simulated page 5:

“This package generates three omics datasets with imposed reference clustering. Starting from real omics (DNA methylation, transcriptome and protein expression) extracted from TCGA ovarian cancer datasets, InterSIM generates clusters and associates features to these clusters by shifting their mean values by a fixed amount. InterSIM preserves the covariance matrix between all pairs of omics and thereby maintains realistic inter- and intra-omics relationships. Importantly, we selected this approach to avoid making assumptions on the distribution of the data. Indeed, alternative simulation approaches assume specific sample distributions (e.g. Gaussian in¹⁴). Assuming a Gaussian distribution, for instance, would favor jDR methods that also make the assumption of Gaussian sample distribution.”

7) Lines 206, not clear if the choice of clustering approach has a strong influence on the results. Please include other common clustering approaches, hierarchical (with average/ward linkage) and a simple k-means.

Thanks for this comment, as we had omitted the name of the clustering algorithm in the result section. We apply simple k-means for clustering the factor matrix F. Indeed, i) we wanted to use a common and simple approach, ii) k-means is the clustering algorithm used in previous evaluations of the clustering performance of jDR [1,2], and iii) the two jDR algorithms benchmarked in our work that provide, as part of their output, the sample clustering (iCluster and intNMF) also use k-means. Thus, to fairly compare all the jDR methods (clustering and non-clustering algorithms), we decided to use k-means also on the other jDR approaches.

This information was previously provided in the methods section “**Clustering of factor matrix**” (page 15), but we now added an explicit mention of the k-mean algorithm in the Results section (page 6), as well as in Figure 2, containing a schematic representation of our analysis protocol on simulated data.

References:

1. Chauvel, Cécile, et al. "Evaluation of integrative clustering methods for the analysis of multi-omics data." *Briefings in Bioinformatics* 21.2 (2020): 541-552
2. Pierre-Jean, Morgane, et al. "Clustering and variable selection evaluation of 13 unsupervised methods for multi-omics data integration." *Briefings in Bioinformatics* (2019).

8) To evaluate clustering performance, more robust metrics have been proposed. Please use the Adjusted Rand Index.

This is an interesting point. We had selected the Jaccard index (JI) because it provides a quality score for each cluster. The Adjusted Rand Index (ARI) is different as it provides one score for the complete set of clusters of each approach. However, these two scores are interesting and complementary for the evaluations, and we now have added the ARI scores to all our clustering experiments (Figures 2, Figure 6, and Supplementary Figure 2). The definition of ARI has been also added in the methods (page 16).

For example, in Figure 2, we added the ARI values in parenthesis after each method name on the x axis. Importantly, this additional score also points to the same 3 best-performing methods (RGGCA, MCIA, MOFA).

9) Line 220: For practical purposes, results with side information should be provided to show if including additional information improve the results, given that this approach can leverage this additional information.

As the goal of our work is to compare different jDR methods, the use of additional information would bias the results of our benchmark towards the only method able to leverage this information (Sckit-fusion). In addition, using side information would open to additional questions, as the use of different side information (e.g. Gene Ontology, PPI network...) could impact differently the results. Overall, this would require an adapted and proper benchmark. For these reasons, we think that considering side information is outside of the scope of the current momix benchmark. But of course, the Jupyter notebook could be used as a basis for further experiments in this direction.

10) Lines 232-233. What about using the subtypes defined by papers published by the TCGA consortium?

Indeed, in the initial version of the manuscript, we used only the subtypes provided by ER/PR/HER-2 markers as a proxy to evaluate the clustering performances of the jDR approaches on TCGA BRCA dataset. As suggested by the reviewer, we have now extended this analysis by comparing the jDR clusters with the subtypes defined in the TCGA consortium paper, namely the Cluster of Clusters (COCA) [1].

These new results are reported in the **Supplementary Figure 1**. In addition, we added the following text to detail this complementary analysis in the results section (see pages 6-7):

“Importantly, we do not have ground-truth cancer subtypes to evaluate the performances of the jDR methods. However, in Breast cancer, we compared the jDR clustering results with two subtypings: the ER/PR/HER-2 subtyping based on Estrogen Receptor (ER), Progesterone Receptor (PR) and HER-2 immunohistochemistry markers²⁵, and the Cluster of Cluster Assignment (COCA) integrative classification performed by the TCGA consortium²⁶. Both subtypings cannot be considered as ground-truth for evaluating jDR clustering performances. The ER/PR/HER-2 overlaps with the PAM50 subtyping, which is obtained using only transcriptomics data, and composed of four subtypes: Basal, Her2, Luminal A and Luminal B²⁷. The COCA subtyping is integrative but has been obtained by separately clustering

different omics and then performing a consensus of the obtained results. Thereby, it does not take into account the complementarity of the various omics.

We decomposed the multi-omics Breast cancer datasets in four factors, and used the Jaccard Index (JI) and Adjusted Rand Index (ARI) to evaluate the overlap between the clustering obtained from these four factors and the Breast cancer subtypings ER/PR/HER-2 and COCA (Supplementary Figure 1). Most of the methods display low JI ([0.2;0.6]) and ARI ([0.2;0.5]) values. JIVE shows the best performances according to both JI and ARI (JI=0.4, ARI=0.4). MCIA has the best performances according to ARI (ARI=0.5), and intNMF has good performances according to JI, but with high variability ([0.2;0.8]), which results in a low ARI value (ARI=0.28)."

References:

1. Cancer Genome Atlas Network. Comprehensive molecular portraits of human breast tumours. *Nature* **490**, 61–70 (2012)
2. Parker, J. S. *et al.* Supervised risk predictor of breast cancer based on intrinsic subtypes. *J. Clin. Oncol. Off. J. Am. Soc. Clin. Oncol.* **27**, 1160–1167 (2009)

11) Lines 241-244 The low performance of all the methods, suggest that the simulation doesn't recapitulate well real case scenarios.

As described in the previous comments, in the simulations, the clusters represent a real ground-truth, as they have been imposed inside the data. The results of the different jDR approaches are in this context, very good. In the clustering of the real breast cancer data, we do not have ground-truth, and used ER/PR/HER-2 (and now also COCA) as proxies representative of the current knowledge. Thus, low performance might reflect the limitations of these reference subtypings. Overall, the optimal definition of breast cancer subtypes is still an open challenge.

We have now clarified this point in the manuscript (see modifications reported in our answer to question 10 above).

12) It would be nice to show for the factors recovered and associated with survival/clinical annotations/biological processes and pathways what are the genes (or other features) associated and discuss if they have any biological relevance. Also, it is necessary to discuss how to recover important features based on these factors in each of the omic datasets used before the integration.

Concerning the second point "how to recover important features", we now better detail the process to extract markers/features from the factors in the manuscript (page 4):

"The factor matrix (F) can be used to cluster samples, while the columns of the weight matrices (A^i) can be used to extract markers by selecting the top-ranked genes, or to identify pathways by applying pre-ranked GSEA (see¹⁰ for further details)."

Then, the genes/features associated with the factors relevant to survival or other annotations might indeed be of interest for further biological interpretation of specific results. As an

example, we analyzed in-depth the RGCCA factor 6 in breast cancer, which is significantly associated with survival. The top-ranked genes contributing to this factor are presented in the table below. These genes are all involved in processes related to cancer aggressiveness, such as invasion, ECM remodelling or cell migration. In addition, applying pre-ranked GSEA to the same factor, we identified enriched pathways, and confirmed that the majority of these genes are involved in the coordinated regulation of specific cancer-related pathways.

We feel that such deep interpretations for genes/features associated with factors of interest for all jDR methods and annotations is out of the scope of the momix benchmark (10 cancers with 10 methods associated with 10 factors leads to hundreds of top-contributing genes to interpret). However, as this suggested analysis is of possible interest for our possible users, we added a “step-by-step” protocol to do such analyses in the Readme of the momix github (<https://github.com/ComputationalSystemsBiology/momix-notebook/blob/master/README.md>).

Marker gene	Function	Reference
TWIST1	Promotes breast cancer invasion	Xu, Yixiang, et al. "Twist1 promotes breast cancer invasion and metastasis by silencing Foxa1 expression." Oncogene 36.8 (2017): 1157-1166.
CLTCL1	Tumor-associated gene	Jiang, Yi-Zhou, et al. "Genomic and transcriptomic landscape of triple-negative breast cancers: subtypes and treatment strategies." Cancer cell 35.3 (2019): 428-440.
COL6A2	Associated to breast cancer recurrence	Fackler, Mary Jo, et al. "Genome-wide methylation analysis identifies genes specific to breast cancer hormone receptor status and risk of recurrence." Cancer research 71.19 (2011): 6195-6207. Fackler, Mary Jo, et al. "Novel methylated biomarkers and a robust assay to detect circulating tumor DNA in metastatic breast cancer." Cancer research 74.8 (2014): 2160-2170.
COPZ2	in signature predicting resistance to chemotherapy in breast cancer	Farmer, Pierre, et al. "A stroma-related gene signature predicts resistance to neoadjuvant chemotherapy in breast cancer." Nature medicine 15.1 (2009): 68.
ELFN2	long non-coding RNA already identified as biomarker of TNBC	Yang, Rui, et al. "Comprehensive analysis of differentially expressed profiles of lncRNAs/mRNAs and miRNAs with associated ceRNA networks in triple-negative breast cancer." Cellular Physiology and Biochemistry 50.2 (2018): 473-488.
FAM171A2	associated to oncogenic RAS response	Blazantin, Nicholas, et al. "ER stress and distinct outputs of the IRE1 α RNase control proliferation and senescence in response to oncogenic Ras." Proceedings of the National Academy of Sciences 114.37 (2017): 9900-9905.
HMX1	TF involved in hypoxia	Hu, Mingxing, et al. "A hypoxia-specific

		and mitochondria-targeted anticancer theranostic agent with high selectivity for cancer cells." Journal of Materials Chemistry B 6.16 (2018): 2413-2416.
KCNJ12	in amplicone associated with HER2+ breast cancer	Arriola, Edurne, et al. "Genomic analysis of the HER2/TOP2A amplicon in breast cancer and breast cancer cell lines." Laboratory investigation 88.5 (2008): 491-503.
MXRA8	ECM remodelling	Lien, HUANG-CHUN, et al. "Molecular signatures of metaplastic carcinoma of the breast by large-scale transcriptional profiling: identification of genes potentially related to epithelial-mesenchymal transition." Oncogene 26.57 (2007): 7859-7871.
SAMD11	In genomic area differentially methylated between breast cancer subtypes	Titus, Alexander J., et al. "Deconvolution of DNA methylation identifies differentially methylated gene regions on 1p36 across breast cancer subtypes." Scientific reports 7.1 (2017): 1-9.
SRPX2	angiogenesis and promoter of cell migration and invasion in multiple cancers	Tanaka, Kaoru, et al. "SRPX2 is overexpressed in gastric cancer and promotes cellular migration and adhesion." International journal of cancer 124.5 (2009): 1072-1080.
ADRA2C	Gene with prognostic significance in breast cancer	Rivero, Ezequiel Mariano, et al. "Prognostic significance of α - and β 2-adrenoceptor gene expression in breast cancer patients." British journal of clinical pharmacology 85.9 (2019): 2143-2154.
SCARF2	Differentially methylated gene in multiple cancers	Yegnasubramanian, Srinivasan, et al. "Chromosome-wide mapping of DNA methylation patterns in normal and malignant prostate cells reveals pervasive methylation of gene-associated and conserved intergenic sequences." BMC genomics 12.1 (2011): 313.

13) Lines 322 324 In addition,... Explain the rationale of this statement and cite relevant work that support this claim.

The rationale of this statement is that by simultaneously analysing multiple single-cell omics, we can use the complementarity of the various omics to compensate for missing or unreliable information present in one omics dataset. For example, additional single-cell omics can recover the missing values of scRNAseq ("dropouts"), as mentioned in Ma et al. (2020). In order to clarify this point, we refined the statement (see page 8) and added references supporting our claim.

"jDR approaches are expected to compensate for the strong intrinsic limitations of single-cell multi-omics, such as small number of sequencing reads, systematic noise due to the stochasticity of gene expression at single-cell level, or dropouts²⁹⁻³¹."

The following references supporting our claim are now cited in the manuscript as 29-31:

1. Hu, Youjin, et al. "Single cell multi-omics technology: methodology and application." *Frontiers in cell and developmental biology* 6 (2018): 28.
2. Stuart, Tim, et al. "Comprehensive integration of single-cell data." *Cell* 177.7 (2019): 1888-1902.
3. Ma, Anjun, et al. "Integrative methods and practical challenges for single-cell multi-omics." *Trends in Biotechnology* (2020).

14) The section 4 needs to be significantly expanded. It doesn't provide in the current form any useful information. Current experimental multi-omics approaches and computational methods for their analysis currently used in the single cell field are marginally described. Please, take a look at this paper that systematically dissect this point: [https://www.cell.com/trends/biotechnology/fulltext/S0167-7799\(20\)30057-3?rss=yes](https://www.cell.com/trends/biotechnology/fulltext/S0167-7799(20)30057-3?rss=yes)

We showed that jDR methods perform well to cluster single-cell data, although these methods were not developed to this goal. We think that this is a relevant and important result. However, to address the Reviewer's concern, we have extended section 4 to cover additional methods, as described below.

We carefully checked the paper proposed by the Reviewer (Ma et al. 2020). This publication is a review focusing on the existing tools for the integrative analysis of scMulti-omics data, going from scRNAseq to scATACseq and further spatial transcriptomics. Some study-cases are presented (Figure 4B), in which the performances of MOFA, Seurat and LIGER are compared regarding the clustering of matched vs unmatched scRNAseq and scATACseq data from lung cancer cells. However, we were not able to find the code associated with these tests, nor any indication about how to download the input lung cancer single-cell data used in the paper. In fact, Ma et al. propose a review of the existing methods rather than a benchmark, and this might be the reason why the information to reproduce independently the results of the paper is not provided. In our case, our goal is to propose an accessible jDR benchmark, enforcing full reproducibility. In this respect, we provide access to the input data, as well as to the code used for all analyses, which is integrated in Jupyter notebooks together with stepwise explanations.

In our manuscript, our main focus is to benchmark a representative collection of jDR methods designed for multi-omics integration. To our knowledge, apart from MOFA, all the existing jDR methods have been only applied to bulk data. The only single-cell specific jDR method is LIGER, which was published just after we had completed the first version of our benchmark. Describing the various existing single-cell omics technologies and the general computational methods for single-cell data analysis thus appear out of the scope of this manuscript.

The initial goal of section 4 was to test if jDR methods designed for bulk could perform reasonably well on single-cell data. However, we agree that single-cell integrative approaches are becoming paramount and an extension of section 4 of our manuscript would be of interest for our readers. Following the reviewer's suggestion, we extended our benchmark to include two state-of-the-art methods selected by Ma et al. (2020), namely LIGER and Seurat.

15) Line 337, why do you use the C-index? Before you were using the Jaccard coefficient. I would suggest using as I proposed before the Adjusted Rand Index.

In section 4, we were initially looking at how well the first two factors of the jDR methods can separate cells coming from different cell lines, which is different from comparing the jDR clustering to a reference clustering. In this case, neither Jaccard Index nor Adjusted Rand Index is adapted. This was the reason for using the C-index.

We now have extended our work on single-cell analysis, and included LIGER and Seurat. Contrary to all the other methods, Seurat is not a proper jDR and does not output factors. Seurat is thus not readily comparable with the other jDRs. In order to still perform comparisons, we additionally applied a k-means clustering of the cells from the outputs of all jDR methods previously considered, plus Seurat and LIGER, and used the JI and ARI scores to compare their performance. The explanation of the clustering evaluation performed on single-cell multi-omics data has been extended in the methods section “**Quality of single-cell clusters**”. The results of this analysis are presented in Figure 6B of the manuscript and in this answer to the next comment.

16) Also approaches specifically developed for single cell data are not tested, e.g. Conos, Seurat, Liger and MOFA+ should be tested. See again the paper mentioned in 14).

Ma et al. 2020 considered Seurat, LIGER and MOFA. As described in the previous comments, we now have extended our section 4 to include Seurat and LIGER, two state-of-the-art methods for multi-omics single-cell integration. Please note that we did not consider MOFA+ that, as compared to MOFA, has the additional capacity to include all the features into the model and work with matrices containing blocks. However, the mathematical models behind the two MOFA versions are the same, and would have resulted in the same performance in our benchmark. In addition, we excluded Conos because, apart from a sparse description in a supplementary note, it is mainly presented as a batch correction tool. Conos requires building a similarity network between cells. This strictly requires to map the ATAC-seq peaks on their corresponding genes, which is a task not relevant for our study.

As mentioned previously, we performed a new analysis and clustered cells from the output of all jDR methods plus Seurat and LIGER, and used the JI and ARI to compare their performances. In this new benchmark, MCIA, tICA and MSFA performed better than both

LIGER and Seurat (new Figure 6B).

We added a section describing these new results page 9:

“ To further compare the performances of jDR approaches with state-of-the-art single-cell multi-omics integrative tools, we further included in our analysis Seurat³³ and LIGER³⁴. Importantly, Seurat does not output factors as the other methods. We thus compared the methods based on their clustering abilities following the same procedure as in the simulation benchmark (Figure 6B). Strikingly, although initially not designed for single-cell data analysis, jDR methods perform equally well or better than Seurat and LIGER. Overall, with MCIA, tICA and MSFA were the best performing algorithms. ”

17) Line 418-420 for the methods presented it is necessary to explore what was recovered in other omics-specific factor and explain to the reader how to potentially interpret this additional information.

The different omics data are expected to contain shared and omic-specific information. However, not all methods can take into account both shared and omic-specific information: RGCCA, MCIA, JIVE and MOFA are the only methods presented in this benchmark that have this capacity.

To provide an example of the additional information provided by omic-specific factors, we considered the RGCCA omics-specific factors relative to methylation and microRNAs in addition to the transcriptomics omics-specific factors considered previously. We tested their association with survival. The ten TCGA cancer types led to a total of 300 factors. The factors that are significantly associated with survival are 15 miRNA-specific, 16 methylation-specific and 11 transcriptomics-specific. Of particular interest, in ovarian cancer, in which no jDR method was able to detect survival-associated factors, the RGCCA factor matrix specific to microRNAs can predict survival with a P-value of 10^{-3} .

We have extended the discussion to better explain how the reader can use and interpret this additional information:

“In addition, when using algorithms producing omics-specific factors, we have only evaluated the transcriptome-associated omics-specific factors (Methods). However, the outputs of these methods can often contain additional relevant information in other omics-specific factors. As an example, in our benchmark, the jDR methods did not retrieve factors significantly associated with survival in ovarian cancer. However, a significant association is retrieved with one miRNA-specific factor of RGCCA.”

18) A key point of this paper is to show how different methods perform for data integration. However, how the transcriptome associated factors would change excluding additional omics datasets? Is it true that the performance of these integrative approaches truly benefit of the data integration? This should be shown and discussed. You should have as a baseline DR methods based only on transcriptomic data e.g. NMF, ICA or PCA.

Rapoport et al. (NAR 2018) showed that in most of their tests multi-omics integration is better than transcriptomics analysis (Rappoport, Nimrod, and Ron Shamir. "Multi-omic and multi-view clustering algorithms: review and cancer benchmark." *Nucleic acids research* 46.20 (2018): 10546-10562.).

To further test if multi-omics is more informative than transcriptomics alone, we focused on survival analyses and included PCA into our comparison across 10 TCGA cancers. PCA finds factors significantly associated with survival for 5 out of 10 cancers, and thus shows lower performances than MCIA, RGCCA, and JIVE (which obtained significant factors for 7 out of 10 cancers). The results have been included in the paper as Supplementary Figure 3.

“It is to note that these jDR methods are also the best performing when compared to DR applied to transcriptome alone (Supplementary Figure 3).”

19) Line 430, please explain “slightly non linear signal”, which dataset are you referring to?

We are here referring to the single-cell multi-omics data presented in the MOFA paper [1]. The dataset is composed of 87 mouse embryonic stem cells (mESCs), composed of 16 cells cultured in “2i” media, which induces a naive pluripotency state, and 71 cells cultured in serum, poised for cellular differentiation [1]. All cells were profiled using single-cell methylation and transcriptome sequencing. As shown in Figure 5E Argelaguet et al. [1], MOFA applied to these data could reconstruct differentiation trajectory from naive pluripotent cells via primed pluripotent cells to differentiated cells. MOFA, despite being a linear method, was able to detect such nonlinear trajectory. We now further specified this point in the manuscript (page 11):

“MOFA is the only algorithm in our benchmark that can also detect slightly nonlinear signals, as shown in¹⁵ for mouse embryonic stem cells.”

References:

1. Argelaguet, Ricard, et al. "Multi-Omics Factor Analysis — a framework for unsupervised integration of multi-omics data sets." *Molecular systems biology* 14.6 (2018).

Reviewers' comments:

Reviewer #1 (Remarks to the Author):

I thank the authors for their extensive reply and the suggested literature. However, my main concerns remain as detailed below.

Major points

1) The biggest concern is lack of conceptual advance. This paper takes 9 previously published methods and compares them against a number of tasks. In my opinion such a benchmarking only exercise simply lacks innovation or novelty that I would expect from a manuscript published in Nature Communications, whose Aims & Scope states "Papers published by the journal aim to represent important advances of significance to specialists within each field." Based on that alone, I think this work is better suited for another journal.

2) I am sorry if my comments misled the authors to believe that my comments are based on confusing "dimensionality reduction applied to single omics for data visualization and the use of dimensionality reduction for multi-omics integration." I am well aware of the differences, but in practice visualisation and data integration are often interconnected. In fact, the cluster analysis shown in Fig. 6 of this paper is a good example. Also, in their rebuttal the authors say: "On one side, dimensionality reduction approaches are used for omics data visualization (i.e., using approaches such as PCA, ICA, NMF, tSNE, or UMAP) [1]. However, the benchmark proposed here concerns another purpose and thus a different set of tools, dedicated to the dimensionality reduction for multiple omics data integration." But then, they use tools like intNMF which is an extension of non-Negative Matrix Factorization (NMF); tICA which is an extension of Independent Component Analysis (ICA), and MCIA and JIVE which are different extensions of Principal Component Analysis (PCA). Now, I admit that I am confused.

3) The technical limitations, which I raised are largely not addressed in the revised version. My comments were mainly aimed at including also non-linear dimensionality reduction methods, which are a hot topic and likely more suitable for biological data that are often non-linear. In their rebuttal the authors argue that there are no examples for that. It is true that this is rather new, and hence would have added novelty. However, t-SNE, for instance, has been used for both data visualization and multiomics data integration {Mehtonen, 2019 #67}. The suggestion of using non-linear or state-of-the-art data dimensionality reduction techniques was meant as suggestions to introduce novelty and true scientific advancement into the paper. In the rebuttal the authors state that they only found applications of variational autoencoders and RBM to digits and image reconstruction. This is just an application of a well establish method that solves the dimensionality reduction problem. In fact, the authors used methods based on PCA, ICA, NMF etc. all of which were originally not designed for omics data. In the original papers of these algorithms they were applied to other fields. The mathematical formulation behind RBMs and autoencoders is in fact parallel to the formulation of the algorithms the authors used. However, they have the advantage of using non-linear transformations and that they can be applied as deep models. The suggestion was for the authors to simply input their data into the algorithm and compare the results. Also, I did not suggest the authors try regression or classification methods such as random forests. I just used them as an example for a successful algorithm that uses non-linear patterns in the data.

Second, I understand that dimensionality reduction can show relationships with underlying biological processes. I am also aware of many instances where this happened, such as the papers

that the authors added. However, it is an entirely different question whether this represents a valid form of comparing the performance of these algorithms. As an example, reproducing the results of the breast cancer molecular subtypes depends on the selection of a list of a few hundred “intrinsic gene list” <https://www.pnas.org/content/100/14/8418>. If a dimensionality reduction technique fails to represent these clusters, it might not be because it is a bad dimension reduction algorithm. It might just be that the overall data does not contain these patterns and they are only present in the “intrinsic” genes. This might explain why most algorithms perform poorly on this task. Or why algorithms did not predict survival well for ovarian, lung and colon cancer. This might not reflect a poorly performing algorithm but rather a matter of there only being a few genes which predict survival for these cancers.

Minor Points:

Page 2 “By integrating multiple sources of omics data, we can reduce the effect of experimental and biological noise.” ... “Indeed, the different omics are complementary, each omics containing information that is not present in others, and multi-omics integration is thereby expected to provide a more comprehensive overview of the biological system.” Statements like these should be referenced and qualified rather than presented as universal truth.

Page 4, 3rd Paragraph “Seven of the nine jDR approaches are extensions of DR methods initially designed for single omics datasets: intNMF is an extension of non-Negative Matrix Factorization (NMF); tICA is an extension of Independent Component Analysis (ICA); MCIA and JIVE are different extensions of Principal Component Analysis (PCA); and MOFA, MSFA, and iCluster are extensions of Factor Analysis.” This line assumes that these dimensionality reduction methods were designed for omics datasets. This is incorrect. PCA was developed in 1901 by Karl Pearson, which somewhat preceded omics data. Similarly, ICA and NMF, were developed originally for dimensionality reduction for other datasets. In particular, the original paper of NMF used facial images and text analysis <https://www.nature.com/articles/44565>.

Page 8, 1st Paragraph. “An optimal jDR method should maximize the number of metagenes enriched in at least one biological annotation, while optimizing also the selectivity (defined as above for clinical annotations and in the Methods).” This line seems subjective. The logic behind this should be explained. Why should the factors be selective for something they were not necessarily designed to do? These factors were designed to represent the multi-omics data.

Reviewer #2 (Remarks to the Author):

I think the authors did an excellent job addressing all my concerns and I really like the new figures. I don't have any further concerns. Thank you for providing the additional analyses I was requesting.

Point by point reply to reviewers

Reviewer 1:

I thank the authors for their extensive reply and the suggested literature. However, my main concerns remain as detailed below.

1) The biggest concern is lack of conceptual advance. This paper takes 9 previously published methods and compares them against a number of tasks. In my opinion such a benchmarking only exercise simply lacks innovation or novelty that I would expect from a manuscript published in Nature Communications, whose Aims & Scope states “Papers published by the journal aim to represent important advances of significance to specialists within each field.” Based on that alone, I think this work is better suited for another journal.

Reply: Benchmarking studies are conceptual advances and valuable resources for the scientific community. As a matter of fact, benchmarking studies have been previously published in *Nature Communications*. The most recent benchmarking studies are:

- “Systematic benchmarking of omics computational tools” (<https://www.nature.com/articles/s41467-019-09406-4>),
- “Benchmarking tomographic acquisition schemes for high-resolution structural biology” (<https://www.nature.com/articles/s41467-020-14535-2>).

Benchmarking studies are also regularly published in other high-impact journals of the *Nature* group, for instance:

- In *Nature Methods*: “Benchmarking algorithms for gene regulatory network inference from single-cell transcriptomic data” (<https://www.nature.com/articles/s41592-019-0690-6>)
- In *Nature biotechnology*: “A comparison of single-cell trajectory inference methods” (<https://www.nature.com/articles/s41587-019-0071-9?platform=hootsuite>)
- In *Nature biotechnology*: “Benchmarking single-cell RNA-sequencing protocols for cell atlas projects” (<https://www.nature.com/articles/s41587-020-0469-4>)

Genome Biology has dedicated an entire special issue to benchmarks (<https://www.biomedcentral.com/collections/benchmarkingstudies>), where they clearly state “As increasing numbers of methods are published in certain fields, it can be difficult to keep track of best practices for their use. Large scale studies that benchmark these methods on a wide range of datasets can be extremely useful to the scientific community.”

2) I am sorry if my comments misled the authors to believe that my comments are based on confusing “dimensionality reduction applied to single omics for data visualization and the use of dimensionality reduction for multi-omics integration.” I am well aware of the differences, but in practice visualisation and data integration are often interconnected. In fact, the cluster analysis shown in Fig. 6 of this paper is a good example. Also, in their rebuttal the authors say: “On one side, dimensionality reduction approaches are used for omics data visualization (i.e., using approaches such as PCA, ICA, NMF, tSNE, or UMAP) [1]. However, the benchmark proposed here concerns another purpose and thus a different set of tools, dedicated to the dimensionality reduction for multiple omics data integration.” But then, they

use tools like intNMF which is an extension of non-Negative Matrix Factorization (NMF); tICA which is an extension of Independent Component Analysis (ICA), and MCIA and JIVE which are different extensions of Principal Component Analysis (PCA). Now, I admit that I am confused.

Reply: PCA, ICA and NMF were developed in 1900-1990, and since then have been applied to solve a wide range of problems in different fields, going from finance to image analysis and visualisation in the field of (functional) genomics.

In the context of data visualisation, given the high-dimensionality of omics data, PCA/NMF/ICA are applied to reduce the dimension of the data before visualization. For this application PCA/NMF/ICA are used in their original formulation with no modification of the mathematical formulation.

In the context of joint data integration, which is the focus of our paper, some methods have been proposed as extensions of PCA (MCIA and JIVE), ICA (tICA) and NMF (intNMF). To this goal, the mathematical formulations of PCA, ICA and NMF have been redefined to co-factorize multiple matrices at the same time. These extensions produced new methods that cannot be considered as rough applications of the initial PCA/NMF/ICA. As a matter of fact, each of these methods has been published in renowned journals. We thus do not find any contradiction between our two sentences.

3) The technical limitations, which I raised are largely not addressed in the revised version. My comments were mainly aimed at including also non-linear dimensionality reduction methods, which are a hot topic and likely more suitable for biological data that are often non-linear. In their rebuttal the authors argue that there are no examples for that. It is true that this is rather new, and hence would have added novelty. However, t-SNE, for instance, has been used for both data visualization and multiomics data integration {Mehtonen, 2019 #67}.

Reply: We will comment on the inclusion of non-linear dimensionality reduction methods below. In the publication mentioned by the Reviewer {Mehtonen, 2019 #67}, the t-SNE method is not applied for true data integration but used for data visualisation. In our work, we refer to data integration as extracting shared signals from multiple omics datasets. What Mehtonen and colleagues did in their 2019 study was to use tSNE to plot multiple transcriptional datasets together (see figure 3 of Mehtonen, 2019), which is a different and much simpler task.

The suggestion of using non-linear or state-of-the-art data dimensionality reduction techniques was meant as suggestions to introduce novelty and true scientific advancement into the paper.

Reply: As stated above, we disagree with this opinion. A benchmark contains novelty and can be true scientific advancement, for example by pointing to the best performing methods depending on the questions and datasets considered. As detailed below, developing a true non-linear multi-omics jDR is not trivial and out-of-the-scope of our current work, but is an interesting research direction for future work.

In the rebuttal the authors state that they only found applications of variational autoencoders and RBM to digits and image reconstruction. This is just an application of a well established method that solves the dimensionality reduction problem. In fact, the authors used methods based on PCA, ICA, NMF etc. all of which were originally not designed for omics data. In the original papers of these algorithms they were applied to other fields. The mathematical formulation behind RBMs and autoencoders is in fact parallel to the formulation of the algorithms the authors used. However, they have the advantage of using non-linear transformations and that they can be applied as deep models. The suggestion was for the authors to simply input their data into the algorithm and compare the results. Also, I did not suggest the authors try regression or classification methods such as random forests. I just used them as an example for a successful algorithm that uses non-linear patterns in the data.

Reply: Our data cannot simply be given as input to the existing non-linear algorithms (such as autoencoders and RBM) because their mathematical formulations are designed to analyse single datasets. Successfully extending the mathematical formulations of autoencoders and RBM for the joint integration of multiple omics dataset, and implementing corresponding tools, is not trivial but interesting, and would deserve to be reported in individual research articles (as done for the extension of the mathematical formulations of PCA/NMF/ICA for joint multi-omics integration). In addition, the latent data representation of RBM and autoencoders is not readily comparable with the factors that we get from the benchmarked methods. Finally, we were here interested in comparing methods already ready to use (implemented in R or Python), a requirement that is common to many benchmarks (<https://www.nature.com/articles/s41592-019-0690-6>, <https://www.nature.com/articles/s41587-019-0071-9?platform=hootsuite>)

Second, I understand that dimensionality reduction can show relationships with underlying biological processes. I am also aware of many instances where this happened, such as the papers that the authors added. However, it is an entirely different question whether this represents a valid form of comparing the performance of these algorithms. As an example, reproducing the results of the breast cancer molecular subtypes depends on the selection of a list of a few hundred “intrinsic gene list” <https://www.pnas.org/content/100/14/8418>. If a dimensionality reduction technique fails to represent these clusters, it might not be because it is a bad dimension reduction algorithm. It might just be that the overall data does not contain these patterns and they are only present in the “intrinsic” genes. This might explain why most algorithms perform poorly on this task. Or why algorithms did not predict survival well for ovarian, lung and colon cancer. This might not reflect a poorly performing algorithm but rather a matter of there only being a few genes which predict survival for these cancers.

Reply: We disagree with the statement that testing the biological enrichments of the jDR factors is an inaccurate procedure to compare jDR methods. Indeed, the latent factors in which the multi-omics data are decomposed are expected to reflect the main sources of variation present in the multi-omics data. These sources of variation should correspond to the biological processes or clinical/survival effects that are operating in the profiled samples, plus possible technical noise due to the experimental procedures. For instance, a source of variation could be cellular proliferation, because different cancer samples can be associated with different proliferation stages. We thereby consider that an evaluation of jDR methods based on the enriched biological processes or other annotations is relevant. As a matter of fact, we and

others previously published comparisons of DR methods based on their biological enrichment or clinical annotations, see for example (<https://academic.oup.com/bioinformatics/article/35/21/4307/5426054> <https://academic.oup.com/nar/article/46/20/10546/5123392>).

Furthermore, we disagree with the claim that “reproducing the results of the breast cancer molecular subtypes depends on the selection of a list of a few hundred intrinsic gene list”. Indeed, subtypes are sets of patients having similar expression patterns across the full transcriptome, and cancer subtyping is classically performed as an unsupervised task on the full transcriptome dataset (see for example <https://www.nature.com/articles/nm.3175>, <https://www.nature.com/articles/nature11412> <https://journals.plos.org/plosmedicine/article?id=10.1371/journal.pmed.1001453> <https://www.nature.com/articles/nm.3174>).

After performing subtyping on the full dataset, researchers often further provide gene signatures as biomarkers. These gene signatures (which seems to be referred to by the Reviewer as “intrinsic gene list”) are small sets of genes that could be used to re-classify new sets of samples. However, classifying samples based on gene signature is just an approximation intended to help a rapid application of the classification by clinicians, but not the way in which cancer subtyping is obtained beforehand.

Concerning survival, if a few genes would have been predictive of survival in ovarian cancer, we would have found at least one factor having these genes as high contributors (with high weights). In other words, the methods would have identified at least a factor associated with survival. Given that this is not the case, we presume that in these cancers, survival might be associated with other biological information or might be -omics specific.

Minor Points:

Page 2 “By integrating multiple sources of omics data, we can reduce the effect of experimental and biological noise.” ... “Indeed, the different omics are complementary, each omics containing information that is not present in others, and multi-omics integration is thereby expected to provide a more comprehensive overview of the biological system.” Statements like these should be referenced and qualified rather than presented as universal truth.

Reply: These points are discussed in multiple papers; we can add citations to relevant publications, such as:

<https://www.nature.com/articles/s41467-017-02467-3>

<https://bmcbioinformatics.biomedcentral.com/articles/10.1186/s12859-019-3224-4>

<https://www.sciencedirect.com/science/article/abs/pii/S0065266015000516?via%3Dihub>

<https://www.frontiersin.org/articles/10.3389/fgene.2017.00084/full>

<https://www.sciencedirect.com/science/article/pii/S2452310018300039>

Page 4, 3rd Paragraph “Seven of the nine jDR approaches are extensions of DR methods initially designed for single omics datasets: intNMF is an extension of non-Negative Matrix Factorization (NMF); tICA is an extension of Independent Component Analysis (ICA); MCIA and JIVE are different extensions of Principal Component Analysis (PCA); and MOFA, MSFA, and iCluster are extensions of Factor Analysis.” This line assumes that these dimensionality reduction methods were designed for omics datasets. This is incorrect. PCA was developed

in 1901 by Karl Pearson, which somewhat preceded omics data. Similarly, ICA and NMF, were developed originally for dimensionality reduction for other datasets. In particular, the original paper of NMF used facial images and text analysis <https://www.nature.com/articles/44565>.

Reply: We agree with the Reviewer that our sentence could be misleading. We corrected it with “Seven of the nine jDR approaches are extensions of DR methods previously used for single omics datasets”.

Page 8, 1st Paragraph. “An optimal jDR method should maximize the number of metagenes enriched in at least one biological annotation, while optimizing also the selectivity (defined as above for clinical annotations and in the Methods).” This line seems subjective. The logic behind this should be explained. Why should the factors be selective for something they were not necessarily designed to do? These factors were designed to represent the multi-omics data.

Reply: It depends on what “represent the multi-omics data” means for the Reviewer. If he/she means visualizing the data, we disagree. As stated above, the latent factors in which the multi-omics data are decomposed are expected to reflect the main sources of variation present in the multi-omics data. These sources of variation should correspond to the biological processes or clinical/survival effects that are operating in the profiled samples, plus possible technical noise due to the experimental procedures. If a method associates multiple factors/dimensions to the same biological process, it loses the opportunity to identify other processes operating in the profiled samples. As a matter of fact, we previously discussed the selectivity of the DR factors in another publication (<https://academic.oup.com/bioinformatics/article/35/21/4307/5426054>).

Reviewer #2 :

I think the authors did an excellent job addressing all my concerns and I really like the new figures. I don't have any further concerns. Thank you for providing the additional analyses I was requesting.

Reply: We thank the reviewer for his/her kind, supportive comments.

REVIEWERS' COMMENTS

Reviewer #3 (Remarks to the Author):

I have been asked to assess the authors' response to the Reviewer #1's comments. Therefore, I will **not** address the manuscript as a whole, only the points raised by Reviewer #1.

1) Quoting Reviewer #1,

> In my opinion such a benchmarking only exercise simply lacks innovation or novelty that I would expect from a manuscript published in Nature Communications

I disagree with Reviewer #1 on this point. Benchmarks are essential for the progress of the computational biology field. To the best of my knowledge, there are no papers comparing joint Dimensionality Reduction (jDR) methods. Therefore, I consider the authors' contribution to be worthy of Nature Communications.

2) If I understand Reviewer #1 correctly, she or he is confused by the main thesis of the benchmark, which is comparison of jDR methods for multi-omics data sets. I agree with Reviewer #1 that visualization and data integration are interconnected. However, by their definition, a benchmark requires a more discrete view of the methods involved (see Weber et al., Genome Biology 2019). Therefore, I think that the authors have taken a legitimate approach by focusing on the visualizations and metrics provided in the manuscript.

3) Reviewer #1 has asked that the authors include a non-linear dimensionality reduction method, such as t-SNE. To the best of my knowledge, no non-linear jDR methods exist. Therefore, the authors cannot include one in a benchmark, whose purpose is to compare existing methods (not develop new methods).

In addition, I have reviewed Mehtonen et al., Nucleic Acids Research 2019. I agree with the authors that the use of t-SNE there does not constitute jDR but rather separate runs of t-SNE, one for each data set. Therefore, it cannot and should not be included in this manuscript.

Answering specific comments,

> The suggestion was for the authors to simply input their data into the algorithm and compare the results.

where "the algorithm" refers to variational autoencoders or to Restricted Boltzmann machine (RBM), both of which are artificial neural networks that are used in the context of dimensionality reduction. The application of such methods to multi-omics data is non-trivial and is therefore outside the scope of this manuscript.

> I understand that dimensionality reduction can show relationships with underlying biological processes. ... However, it is an entirely different question whether this represents a valid form of

comparing the performance of these algorithms.

The purpose of a benchmark is to compare the performance of a set of algorithms to each other, not against an underlying truth such as a biological process. The authors provide a set of well-defined metrics and visualizations which are used for the comparison. In my opinion, these are reasonable metrics and fulfill the purpose of the benchmark. Therefore, I do not agree with Reviewer #1's decoupling between the algorithm performance and the metrics reported by the authors.

In summary, Reviewer #1 raises three major objections: (1) benchmarking lacks innovation or novelty, (2) confusion around the separation between traditional and multi-omic analysis, and (3) no representation of non-linear dimensionality reduction. It is my opinion that these objections are not relevant to the purpose of the manuscript as a benchmark of jDR methods.

Therefore, in my opinion, Reviewer #1's comments should not hinder the publication of this manuscript.

El-ad David Amir
(He/Him/His)

POINT-BY-POINT ANSWER TO REVIEWERS' COMMENTS

We thank Reviewer 3 for the time dedicated to examine our exchanges with Reviewer 1, and for supporting our claims. You can find below our point-by-point comments.

Reviewer #3 (Remarks to the Author):

I have been asked to assess the authors' response to the Reviewer #1's comments. Therefore, I will *not* address the manuscript as a whole, only the points raised by Reviewer #1.

1) Quoting Reviewer #1,

> In my opinion such a benchmarking only exercise simply lacks innovation or novelty that I would expect from a manuscript published in Nature Communications

I disagree with Reviewer #1 on this point. Benchmarks are essential for the progress of the computational biology field. To the best of my knowledge, there are no papers comparing joint Dimensionality Reduction (jDR) methods. Therefore, I consider the authors' contribution to be worthy of Nature Communications.

Reply: We thank the Reviewer for his/her comment. We also strongly believe in the importance of benchmarks to further improve existing methodologies.

2) If I understand Reviewer #1 correctly, she or he is confused by the main thesis of the benchmark, which is comparison of jDR methods for multi-omics data sets. I agree with Reviewer #1 that visualization and data integration are interconnected. However, by their definition, a benchmark requires a more discrete view of the methods involved (see Weber et al., Genome Biology 2019). Therefore, I think that the authors have taken a legitimate approach by focusing on the visualizations and metrics provided in the manuscript.

Reply: we agree with both Reviewers that visualization and jDR are interconnected. Reviewer 1 insisted on the fact that jDR were mainly used for visualisation. We contrarily wanted to emphasize that visualization is just one of the many applications of jDR, and not the most relevant for benchmarking methods.

3) Reviewer #1 has asked that the authors include a non-linear dimensionality reduction method, such as t-SNE. To the best of my knowledge, no non-linear jDR methods exist. Therefore, the authors cannot include one in a benchmark, whose purpose is to compare existing methods (not develop new methods).

In addition, I have reviewed Mehtonen et al., Nucleic Acids Research 2019. I agree with the authors that the use of t-SNE there does not constitute jDR but rather separate runs of t-SNE, one for each data set. Therefore, it cannot and should not be included in this manuscript.

Reply: We fully agree with this summary of Mehtonen et al. and the Reviewer's interpretation and conclusion.

Answering specific comments,

> The suggestion was for the authors to simply input their data into the algorithm and compare the results.

where "the algorithm" refers to variational autoencoders or to Restricted Boltzmann machine (RBM), both of which are artificial neural networks that are used in the context of dimensionality reduction. The application of such methods to multi-omics data is non-trivial and is therefore outside the scope of this manuscript.

Reply: We fully agree with the Reviewer's comments. Indeed, we are now starting to work on developing non-linear approaches for multi-omics integration.

> I understand that dimensionality reduction can show relationships with underlying biological processes. ... However, it is an entirely different question whether this represents a valid form of comparing the performance of these algorithms.

The purpose of a benchmark is to compare the performance of a set of algorithms to each other, not against an underlying truth such as a biological process. The authors provide a set of well-defined metrics and visualizations which are used for the comparison. In my opinion, these are reasonable metrics and fulfill the purpose of the benchmark. Therefore, I do not agree with Reviewer #1's decoupling between the algorithm performance and the metrics reported by the authors.

Reply: We thank the Reviewer for supporting our claims

In summary, Reviewer #1 raises three major objections: (1) benchmarking lacks innovation or novelty, (2) confusion around the separation between traditional and multi-omic analysis, and (3) no representation of non-linear dimensionality reduction. It is my opinion that these objections are not relevant to the purpose of the manuscript as a benchmark of jDR methods.

Therefore, in my opinion, Reviewer #1's comments should not hinder the publication of this manuscript.